# Finding Mixed Nash Equilibria of Generative Adversarial Networks

## Abstract

We reconsider the training objective of Generative Adversarial Networks (GANs) from the *mixed Nash Equilibria* (NE) perspective. Inspired by the classical prox methods, we develop a novel algorithmic framework for GANs via an infinite-dimensional two-player game and prove rigorous convergence rates to the mixed NE. We then propose a principled procedure to reduce our novel prox methods to simple sampling routines, leading to practically efficient algorithms. Finally, we provide experimental evidence that our approach outperforms methods that seek pure strategy equilibria, such as SGD, Adam, and RMSProp, both in speed and quality.

## 1 Introduction

The Generative Adversarial Network (GAN) (Goodfellow et al., 2014) has become one of the most powerful paradigms in learning real-world distributions, especially for image-related data. It has been successfully applied to a host of applications such as image translation (Isola et al., 2017; Kim et al., 2017; Zhu et al., 2017), super-resolution imaging (Wang et al., 2015), pose editing (Pumarola et al., 2018b), and facial animation (Pumarola et al., 2018a).

Despite of the many accomplishments, the major hurdle blocking the full impact of GAN is its notoriously difficult training phase. In the language of game theory, GAN seeks for a *pure strategy* equilibrium, which is well-known to be ill-posed in many scenarios (Dasgupta & Maskin, 1986). Indeed, it is known that a pure strategy equilibrium might not exist (Arora et al., 2017), might be degenerate (Sønderby et al., 2017), or cannot be reliably reached by existing algorithms (Mescheder et al., 2017).

Empirically, it has also been observed that common algorithms, such as SGD or Adam (Kingma & Ba, 2015), lead to unstable training. While much efforts have been devoted into understanding the training dynamics of GANs (Balduzzi et al., 2018; Gemp & Mahadevan, 2018; Gidel et al., 2018a;b; Liang & Stokes, 2018), a provably convergent algorithm for general GANs, even under reasonably strong assumptions, is still lacking.

In this paper, we address the above problems with the following contributions:

1. We propose to study the *mixed Nash Equilibrium* (NE) of GANs: Instead of searching for an optimal pure strategy which might not even exist, we optimize over the set of *probability distributions* over pure strategies of the networks. The existence of a solution to such problems was long established amongst the earliest game theory work (Glicksberg, 1952), leading to well-posed optimization problems.

2. We demonstrate that the prox methods of (Nemirovsky & Yudin, 1983; Nemirovski, 2004), which are fundamental building blocks for solving two-player games with *finitely* many strategies, can be extended to continuously many strategies, and hence applicable to training GANs. We provide an elementary proof for their convergence rates to learning the mixed NE.

3. We construct a principled procedure to reduce our novel prox methods to certain sampling tasks that were empirically proven easy by recent work (Chaudhari et al., 2017; 2018; Dziugaite & Roy, 2018). We further establish heuristic guidelines to greatly scale down the memory and computational costs, resulting in simple algorithms whose per-iteration complexity is almost as cheap as SGD.

4. We experimentally show that our algorithms consistently achieve better or comparable performance than popular baselines such as SGD, Adam, and RMSProp (Tieleman & Hinton, 2012).

**Related Work:** While the literature on training GANs is vast, to our knowledge, there exist only few papers on the mixed NE perspective. The notion of mixed NE is already present in (Goodfellow et al., 2014), but is stated only as an existential result. The authors of (Arora et al., 2017) advocate the mixed strategies, but do not provide a provably convergent algorithm. (Oliehoek et al., 2018) also considers mixed NE, but only with finitely many parameters. The work (Grnarova et al., 2018) proposes a provably convergent algorithm for finding the mixed NE of GANs under the unrealistic assumption that the discriminator is a single-layered neural network. In contrast, our results are applicable to arbitrary architectures, including popular ones (Arjovsky et al., 2017; Gulrajani et al., 2017).

Due to its fundamental role in game theory, many prox methods have been applied to study the training of GANs (Daskalakis et al., 2018; Gidel et al., 2018a; Mertikopoulos et al., 2018). However, these works focus on the classical pure strategy equilibria and are hence distinct from our problem formulation. In particular, they give rise to drastically different algorithms from ours and do not provide convergence rates for GANs.

In terms of analysis techniques, our framework is closely related to (Balandat et al., 2016), but with several important distinctions. First, the analysis of (Balandat et al., 2016) is based on dual averaging (Nesterov, 2009), while we consider Mirror Descent and also the more sophisticated Mirror-Prox (see Section 3). Second, unlike our work, (Balandat et al., 2016) do not provide any convergence rate for learning mixed NE of two-player games. Finally, (Balandat et al., 2016) is only of theoretical interest with no practical algorithm.

**Notation:** Throughout the paper, we use $\boldsymbol{z}$ to denote a generic variable and $\mathcal{Z} \subseteq \mathbb{R}^d$ its domain. We denote the set of all Borel probability measures on $\mathcal{Z}$ by $\mathcal{M}(\mathcal{Z})$, and the set of all functions on $\mathcal{Z}$ by $\mathcal{F}(\mathcal{Z})$.[1] We write $\mathrm{d}\mu = \rho \mathrm{d}\boldsymbol{z}$ to mean that the density function of $\mu \in \mathcal{M}(\mathcal{Z})$ with respect to the Lebesgue measure is $\rho$. All integrals without specifying the measure are understood to be with respect to Lebesgue. For any objective of the form $\min_{\boldsymbol{x}} \max_{\boldsymbol{y}} F(\boldsymbol{x}, \boldsymbol{y})$, we say that $(\boldsymbol{x}_T, \boldsymbol{y}_T)$ is an $O\left(T^{-1/2}\right)$-NE if $\max_{\boldsymbol{x}, \boldsymbol{y}} \{F(\boldsymbol{x}_T, \boldsymbol{y}) - F(\boldsymbol{x}, \boldsymbol{y}_T)\} = O\left(T^{-1/2}\right)$. Similarly we can define $O\left(T^{-1}\right)$-NE. The symbol $\|\cdot\|_{\mathbb{L}^\infty}$ denotes the $\mathbb{L}^\infty$-norm of functions, and $\|\cdot\|_{\mathrm{TV}}$ denotes the total variation norm of probability measures.

## 2 Problem Formulation

We review standard results in game theory in Section 2.1, whose proof can be found in (Bubeck, 2013a;b;c). Section 2.2 relates training of GANs to the two-player game in Section 2.1, thereby suggesting to generalize the prox methods to infinite dimension.

### 2.1 Preliminary: Prox Methods for Finite Games

Consider the classical formulation of a two-player game with *finitely* many strategies:

$$\min_{\boldsymbol{p} \in \Delta_m} \max_{\boldsymbol{q} \in \Delta_n} \langle \boldsymbol{q}, \boldsymbol{a} \rangle - \langle \boldsymbol{q}, A\boldsymbol{p} \rangle, \tag{1}$$

where $A$ is a payoff matrix, $\boldsymbol{a}$ is a vector, and $\Delta_d := \left\{ \boldsymbol{z} \in \mathbb{R}^d \mid \sum_{i=1}^d z_i = 1 \right\}$ is the probability simplex, representing the *mixed strategies* (i.e., probability distributions) over $d$ pure strategies. A pair $(\boldsymbol{p}_{\mathrm{NE}}, \boldsymbol{q}_{\mathrm{NE}})$ achieving the min-max value in (1) is called a mixed NE.

Assume that the matrix $A$ is too expensive to evaluate whereas the (stochastic) gradients of (1) are easy to obtain. Under such settings, a celebrated algorithm, the so-called **entropic Mirror Descent** (entropic MD), learns an $O\left(T^{-1/2}\right)$-NE: Let $\phi(\boldsymbol{z}) := \sum_{i=1}^d z_i \log z_i$ be the entropy function and $\phi^\star(\boldsymbol{y}) := \log \sum_{i=1}^d e^{y_i} = \sup_{\boldsymbol{z} \in \Delta_d} \{\langle \boldsymbol{z}, \boldsymbol{y} \rangle - \phi(\boldsymbol{z})\}$ be its Fenchel dual.

---

[1]Strictly speaking, our derivation requires mild regularity (see Appendix A.1) assumptions on the probability measure and function classes, which are met by most practical applications.

For a learning rate $\eta$ and an arbitrary vector $\boldsymbol{b} \in \mathbb{R}^d$, define the MD iterates as

$$\boldsymbol{z}' = \mathrm{MD}_\eta\left(\boldsymbol{z}, \boldsymbol{b}\right) \quad \equiv \quad \boldsymbol{z}' = \nabla\phi^\star\left(\nabla\phi(\boldsymbol{z}) - \eta\boldsymbol{b}\right) \quad \equiv \quad z_i' = \frac{z_i e^{-\eta b_i}}{\sum_{i=1}^d z_i e^{-\eta b_i}}, \ \ \forall 1 \le i \le d. \quad (2)$$

The equivalence of the last two formulas in (2) can be readily checked.

Denote by $\bar{\boldsymbol{p}}_T := \frac{1}{T}\sum_{t=1}^T \boldsymbol{p}_t$ and $\bar{\boldsymbol{q}}_T := \frac{1}{T}\sum_{t=1}^T \boldsymbol{q}_t$ the ergodic average of two sequences $\{\boldsymbol{p}_t\}_{t=1}^T$ and $\{\boldsymbol{q}_t\}_{t=1}^T$. Then, with a properly chosen step-size $\eta$, we have

$$\begin{cases} \boldsymbol{p}_{t+1} = \mathrm{MD}_\eta\left(\boldsymbol{p}_t, -A^\top\boldsymbol{q}_t\right) \\ \boldsymbol{q}_{t+1} = \mathrm{MD}_\eta\left(\boldsymbol{q}_t, -\boldsymbol{a} + A\boldsymbol{p}_t\right) \end{cases} \Rightarrow \quad (\bar{\boldsymbol{p}}_T, \bar{\boldsymbol{q}}_T) \text{ is an } O\left(T^{-1/2}\right)\text{-NE.}$$

Moreover, a slightly more complicated algorithm, called the **entropic Mirror-Prox** (entropy MP) (Nemirovski, 2004), achieves faster rate than the entropic MD:

$$\begin{cases} \boldsymbol{p}_t = \mathrm{MD}_\eta\left(\tilde{\boldsymbol{p}}_t, -A^\top\tilde{\boldsymbol{q}}_t\right), \quad \tilde{\boldsymbol{p}}_{t+1} = \mathrm{MD}_\eta\left(\tilde{\boldsymbol{p}}_t, -A^\top\boldsymbol{q}_t\right) \\ \boldsymbol{q}_t = \mathrm{MD}_\eta\left(\tilde{\boldsymbol{q}}_t, -\boldsymbol{a} + A\tilde{\boldsymbol{p}}_t\right), \quad \tilde{\boldsymbol{q}}_{t+1} = \mathrm{MD}_\eta\left(\tilde{\boldsymbol{q}}_t, -\boldsymbol{a} + A\boldsymbol{p}_t\right) \end{cases} \Rightarrow \quad (\bar{\boldsymbol{p}}_T, \bar{\boldsymbol{q}}_T) \text{ is an } O\left(T^{-1}\right)\text{-NE.}$$

If, instead of deterministic gradients, one uses unbiased stochastic gradients for entropic MD and MP, then both algorithms achieve $O\left(T^{-1/2}\right)$-NE in expectation.

## 2.2 Mixed Strategy Formulation for Generative Adversarial Networks

For illustration, let us focus on the Wasserstein GAN (Arjovsky et al., 2017), and we perform a common bilinearization trick that dates back at least to the early game theory literature (Glicksberg, 1952), and is also well-known in optimal transport theory (Villani, 2003).

The training objective of Wasserstein GAN is

$$\min_{\boldsymbol{\theta}\in\Theta} \max_{\boldsymbol{w}\in\mathcal{W}} \mathbb{E}_{X\sim\mathbb{P}_{\mathrm{real}}}[f_{\boldsymbol{w}}(X)] - \mathbb{E}_{X\sim\mathbb{P}_{\boldsymbol{\theta}}}[f_{\boldsymbol{w}}(X)], \quad (3)$$

where $\Theta$ is the set of parameters for the generator and $\mathcal{W}$ the set of parameters for the discriminator $f$, typically both taken to be neural nets. As mentioned in the introduction, such an optimization problem can be ill-posed, which is also supported by empirical evidence.

The high-level idea of our approach is, instead of solving (3) directly, we focus on the *mixed strategy* formulation of (3). In other words, we consider the set of all probability distributions over $\Theta$ and $\mathcal{W}$, and we search for the optimal distribution that solves the following program:

$$\min_{\nu\in\mathcal{M}(\Theta)} \max_{\mu\in\mathcal{M}(\mathcal{W})} \mathbb{E}_{\boldsymbol{w}\sim\mu}\mathbb{E}_{X\sim\mathbb{P}_{\mathrm{real}}}[f_{\boldsymbol{w}}(X)] - \mathbb{E}_{\boldsymbol{w}\sim\mu}\mathbb{E}_{\boldsymbol{\theta}\sim\nu}\mathbb{E}_{X\sim\mathbb{P}_{\boldsymbol{\theta}}}[f_{\boldsymbol{w}}(X)]. \quad (4)$$

Define the function $g : \mathcal{W} \to \mathbb{R}$ by $g(\boldsymbol{w}) := \mathbb{E}_{X\sim\mathbb{P}_{\mathrm{real}}}[f_{\boldsymbol{w}}(X)]$ and the operator $G : \mathcal{M}(\Theta) \to \mathcal{F}(\mathcal{W})$ as $(G\nu)(\boldsymbol{w}) := \mathbb{E}_{\boldsymbol{\theta}\sim\nu, \mathbf{X}\sim\mathbb{P}_{\boldsymbol{\theta}}}[f_{\boldsymbol{w}}(X)]$. Denoting $\langle\mu, h\rangle := \mathbb{E}_\mu h^2$ for any probability measure $\mu$ and function $h$, we may rewrite (4) as

$$\min_{\nu\in\mathcal{M}(\Theta)} \max_{\mu\in\mathcal{M}(\mathcal{W})} \langle\mu, g\rangle - \langle\mu, G\nu\rangle. \quad (5)$$

Furthermore, the Fréchet derivative (the analogue of gradient in infinite dimension) of (5) with respect to $\mu$ is simply $g - G\nu$, and the derivative of (5) with respect to $\nu$ is $-G^\dagger\mu$, where $G^\dagger : \mathcal{M}(\mathcal{W}) \to \mathcal{F}(\Theta)$ is the adjoint operator of $G$ defined via the relation

$$\forall\mu\in\mathcal{M}(\mathcal{W}), \nu\in\mathcal{M}(\Theta), \quad \langle\mu, G\nu\rangle = \langle\nu, G^\dagger\mu\rangle. \quad (6)$$

One can easily check that $(G^\dagger\mu)(\boldsymbol{\theta}) := \mathbb{E}_{X\sim\mathbb{P}_{\boldsymbol{\theta}}, \boldsymbol{w}\sim\mu}[f_{\boldsymbol{w}}(X)]$ achieves the equality in (6).

To summarize, the mixed strategy formulation of Wasserstein GAN is (5), whose derivatives can be expressed in terms of $g$ and $G$. We now make the crucial observation that (5) is the infinite-dimensional analogue of (1): The distributions over finite strategies are replaced with probability measures over a continuous parameter set, the vector $\boldsymbol{a}$ is replaced with a function $g$, the matrix $A$ is replaced with a linear operator[3] $G$, and the gradients are replaced with Fréchet derivatives. Based on Section 2.1, it is then natural to ask:

---

[2]It should be noted that $\langle\mu, h\rangle$ is NOT an inner product, and rather is the dual pairing in Banach spaces (Halmos, 2013).

[3]The linearity of $G$ trivially follows from the linearity of expectation.

> *Can the entropic Mirror Descent and Mirror-Prox be extended to infinite dimension to solve (5)? Can we retain the convergence rates?*

We provide an affirmative answer to both questions in the next section.

*Remark.* The derivation in Section 2.2 can be applied to any GAN objective.

## 3 Infinite-Dimensional Prox Methods

This section builds a rigorous infinite-dimensional formalism in parallel to the finite-dimensional prox methods and proves their convergence rates. While simple in retrospect, to our knowledge, these results are new.

### 3.1 Preparation: The Mirror Descent Iterates

We first recall the notion of (Fréchet) derivative in infinite-dimensional spaces. A (nonlinear) functional $\Phi : \mathcal{M}(\mathcal{Z}) \to \mathbb{R}$ is said to possess a derivative at $\mu \in \mathcal{M}(\mathcal{Z})$ if there exists a function $\mathrm{d}\Phi(\mu) \in \mathcal{F}(\mathcal{Z})$ such that, for all $\mu' \in \mathcal{M}(\mathcal{Z})$, we have

$$\Phi(\mu + \epsilon\mu') = \Phi(\mu) + \epsilon \langle \mu', \mathrm{d}\Phi(\mu) \rangle + o(\epsilon).$$

Similarly, a (nonlinear) functional $\Phi^\star : \mathcal{F}(\mathcal{Z}) \to \mathbb{R}$ is said to possess a derivative at $h \in \mathcal{F}(\mathcal{Z})$ if there exists a measure $\mathrm{d}\Phi^\star(h) \in \mathcal{M}(\mathcal{Z})$ such that, for all $h' \in \mathcal{F}(\mathcal{Z})$, we have

$$\Phi^\star(h + \epsilon h') = \Phi^\star(h) + \epsilon \langle \mathrm{d}\Phi^\star(h), h' \rangle + o(\epsilon).$$

The most important functionals in this paper are the (negative) Shannon entropy

$$\mu \in \mathcal{M}(\mathcal{Z}), \quad \Phi(\mu) := \int \mathrm{d}\mu \log \frac{\mathrm{d}\mu}{\mathrm{d}\boldsymbol{z}}$$

and its Fenchel dual

$$h \in \mathcal{F}(\mathcal{Z}), \quad \Phi^\star(h) := \log \int e^h \mathrm{d}\boldsymbol{z}.$$

The first result of our paper is to show that, in direct analogy to (2), the infinite-dimensional MD iterates can be expressed as:

**Theorem 1** (Infinite-Dimensional Mirror Descent, informal)**.** *For a learning rate $\eta$ and an arbitrary function $h$, we can equivalently define*

$$\mu_+ = \mathrm{MD}_\eta(\mu, h) \quad \equiv \quad \mu_+ = \mathrm{d}\Phi^\star(\mathrm{d}\Phi(\mu) - \eta h) \equiv \quad \mathrm{d}\mu_+ = \frac{e^{-\eta h}\mathrm{d}\mu}{\int e^{-\eta h}\mathrm{d}\mu}. \tag{7}$$

*Moreover, most the essential ingredients in the analysis of finite-dimensional prox methods can be generalized to infinite dimension.*

See **Theorem 4** of Appendix A for precise statements and a long list of "essential ingredients of prox methods" generalizable to infinite dimension.

### 3.2 Infinite-Dimensional Prox Methods and Convergence Rates

Armed with results in Section 3.1, we now introduce two "conceptual" algorithms for solving the mixed NE of Wasserstein GANs: The infinite-dimensional entropic MD in **Algorithm 1** and MP in **Algorithm 2**. These algorithms iterate over probability measures and cannot be directly used in practice, but they possess rigorous convergence rates, and hence motivate the reduction procedure in Section 4 to come.

---

**Algorithm 1:** Infinite-Dimensional Entropic MD

**Input:** Initial distributions $\mu_1, \nu_1$, learning rate $\eta$
**for** $t = 1, 2, \ldots, T-1$ **do**
$\quad \lfloor \ \nu_{t+1} = \mathrm{MD}_\eta\left(\nu_t, -G^\dagger\mu_t\right), \quad \mu_{t+1} = \mathrm{MD}_\eta\left(\mu_t, -g + G\nu_t\right);$
return $\bar{\nu}_T = \frac{1}{T}\sum_{t=1}^T \nu_t$ and $\bar{\mu}_T = \frac{1}{T}\sum_{t=1}^T \mu_t.$

---

---

**Algorithm 2:** INFINITE-DIMENSIONAL ENTROPIC MP

---

**Input:** Initial distributions $\tilde{\mu}_1, \tilde{\nu}_1$, learning rate $\eta$

**for** $t = 1, 2, \ldots, T$ **do**

$\quad \nu_t = \mathrm{MD}_\eta \left( \tilde{\nu}_t, -G^\dagger \tilde{\mu}_t \right), \quad \mu_t = \mathrm{MD}_\eta \left( \tilde{\mu}_t, -g + G\tilde{\nu}_t \right);$

$\quad \tilde{\nu}_{t+1} = \mathrm{MD}_\eta \left( \tilde{\nu}_t, -G^\dagger \mu_t \right), \quad \tilde{\mu}_{t+1} = \mathrm{MD}_\eta \left( \tilde{\mu}_t, -g + G\nu_t \right);$

return $\bar{\nu}_T = \frac{1}{T} \sum_{t=1}^T \nu_t$ and $\bar{\mu}_T = \frac{1}{T} \sum_{t=1}^T \mu_t$.

---

**Theorem 2** (Convergence Rates). *Let $\Phi(\mu) = \int \mathrm{d}\mu \log \frac{\mathrm{d}\mu}{\mathrm{d}z}$. Let $M$ be a constant such that $\max \left[ \left\| -g + G\nu \right\|_{\mathbb{L}^\infty}, \left\| G^\dagger \mu \right\|_{\mathbb{L}^\infty} \right] \leq M$, and $L$ be such that $\left\| G(\nu - \nu') \right\|_{\mathbb{L}^\infty} \leq L \left\| \nu - \nu' \right\|_{\mathrm{TV}}$ and $\left\| G^\dagger (\mu - \mu') \right\|_{\mathbb{L}^\infty} \leq L \left\| \mu - \mu' \right\|_{\mathrm{TV}}$. Let $D(\cdot, \cdot)$ be the relative entropy, and denote by $D_0 := D(\mu_{\mathrm{NE}}, \mu_1) + D(\nu_{\mathrm{NE}}, \nu_1)$ the initial distance to the mixed NE. Then*

1. *Assume that we have access to the deterministic derivatives $\left\{ -G^\dagger \mu_t \right\}_{t=1}^T$ and $\left\{ g - G\nu \right\}_{t=1}^T$. Then **Algorithm 1** achieves $O\left( T^{-1/2} \right)$-NE with $\eta = \frac{2}{M} \sqrt{\frac{D_0}{T}}$, and **Algorithm 2** achieves $O\left( T^{-1} \right)$-NE with $\eta = \frac{4}{L}$.*

2. *Assume that we have access to stochastic derivatives $\left\{ -\hat{G}^\dagger \mu_t \right\}_{t=1}^T$ and $\left\{ \hat{g} - \hat{G}\nu \right\}_{t=1}^T$ such that $\max \left[ \mathbb{E} \left\| -\hat{g} + \hat{G}\nu \right\|_{\mathbb{L}^\infty}, \mathbb{E} \left\| \hat{G}^\dagger \mu \right\|_{\mathbb{L}^\infty} \right] \leq M'$, and the variance is upper bounded by $\sigma^2$. Assume also that the bias of stochastic derivatives satisfies $\max \left[ \left\| \mathbb{E}[-\hat{g} + \hat{G}\nu] + g - G\nu \right\|_{\mathbb{L}^\infty}, \left\| \mathbb{E}[\hat{G}^\dagger \mu] - G^\dagger \mu \right\|_{\mathbb{L}^\infty} \right] \leq \tau$. Then **Algorithm 1** with stochastic derivatives achieves $O\left( T^{-1/2} \right)$-NE in expectation with $\eta = \sqrt{\frac{D_0}{T \left( 4\tau + M'/4 \right)}}$, and **Algorithm 2** with stochastic derivatives achieves $\left( O\left( T^{-1/2} \right) + O(\tau) \right)$-NE in expectation with $\eta = \min \left[ \frac{4}{\sqrt{3}L}, \sqrt{\frac{2D_0}{3T\sigma^2}} \right]$.*

The proof can be found in Appendix B and C.

*Remark.* If, as in previous work (Arora et al., 2017), we assume the output of the discriminator to be bounded by $U$, then we have $M, M' \leq 2U$ and $L \leq U$ in **Theorem 2**. The constant error term for stochastic MP is standard; see, e.g., (Juditsky et al., 2011).

## 4 FROM THEORY TO PRACTICE

Section 4.1 reduces **Algorithm 1** and **Algorithm 2** to a sampling routine (Welling & Teh, 2011) that has widely been used in machine learning. Section 4.2 proposes to further simplify the algorithms by summarizing a batch of samples by their mean.

For simplicity, we will only derive the algorithm for entropic MD; the case for entropic MP is similar but requires more computation. To ease the notation, we assume $\eta = 1$ throughout this section as $\eta$ does not play an important role in the derivation below.

### 4.1 IMPLEMENTABLE ENTROPIC MD: FROM PROBABILITY MEASURE TO SAMPLES

Consider **Algorithm 1**. The reduction consists of three steps.

**Step 1: Reformulating Entropic Mirror Descent Iterates**

The definition of the MD iterate (7) relates the updated probability measure $\mu_{t+1}$ to the current probability measure $\mu_t$, but it tells us nothing about the density function of $\mu_{t+1}$, from which we want to sample. Our first step is to express (7) in a more tractable form. By recursively applying (7) and using **Theorem 4.10** in Appendix A, we have, for some

constants $C_1, ..., C_{T-1}$,

$$
\begin{aligned}
\mathrm{d}\Phi(\mu_T) &= \mathrm{d}\Phi(\mu_{T-1}) - (-g + G\nu_{T-1}) + C_{T-1} \\
&= \mathrm{d}\Phi(\mu_{T-2}) - (-g + G\nu_{T-2}) - (-g + G\nu_{T-1}) + C_{T-1} + C_{t-2} \\
&= \cdots = \mathrm{d}\Phi(\mu_1) - \left( -(T-1)g + G\sum_{s=1}^{T-1}\nu_s \right) + \sum_{s=1}^{T-1} C_s.
\end{aligned}
$$

For simplicity, assume that $\mu_1$ is uniform so that $\mathrm{d}\Phi(\mu_1)$ is a constant function. Then, by (13) and that $\mathrm{d}\Phi^\star(\mathrm{d}\Phi(\mu_T)) = \mathrm{d}\mu_T$, we see that the density function of $\mu_T$ is simply $\mathrm{d}\mu_T = \frac{\exp\{(T-1)g - G\sum_{s=1}^{T-1}\nu_s\}\mathrm{d}\boldsymbol{w}}{\int \exp\{(T-1)g - G\sum_{s=1}^{T-1}\nu_s\}\mathrm{d}\boldsymbol{w}}$. Similarly, we have $\mathrm{d}\nu_T = \frac{\exp\{G^\dagger \sum_{s=1}^{T-1}\mu_s\}\mathrm{d}\boldsymbol{\theta}}{\int \exp\{G^\dagger \sum_{s=1}^{T-1}\mu_s\}\mathrm{d}\boldsymbol{\theta}}$.

**Step 2: Empirical Approximation for Stochastic Derivatives**

The derivatives of (5) involve the function $g$ and operator $G$. Recall that $g$ requires taking expectation over the real data distribution, which we do not have access to. A common approach is to replace the true expectation with its empirical average:

$$
g(\boldsymbol{w}) = \mathbb{E}_{X \sim \mathbb{P}_{\mathrm{real}}}[f_{\boldsymbol{w}}(X)] \simeq \frac{1}{n}\sum_{i=1}^{n} f_{\boldsymbol{w}}(X_i^{\mathrm{real}}) \triangleq \hat{g}(\boldsymbol{w})
$$

where $X_i$'s are real data and $n$ is the batch size. Clearly, $\hat{g}$ is an unbiased estimator of $g$.

On the other hand, $G\nu_t$ and $G^\dagger\mu_t$ involve expectation over $\nu_t$ and $\mu_t$, respectively, and also over the fake data distribution $\mathbb{P}_{\boldsymbol{\theta}}$. Therefore, if we are able to draw samples from $\mu_t$ and $\nu_t$, then we can again approximate the expectation via the empirical average:

$$
\boldsymbol{\theta}^{(1)}, \boldsymbol{\theta}^{(2)}, ..., \boldsymbol{\theta}^{(n')} \sim \nu_t, \ \left\{ X_i^{(j)} \right\}_{i=1}^{n} \sim \mathbb{P}_{\boldsymbol{\theta}^{(j)}}, \quad \hat{G}\nu_t(\boldsymbol{w}) \simeq \frac{1}{nn'} \sum_{i=1}^{n}\sum_{j=1}^{n'} f_{\boldsymbol{w}}\left( X_i^{(j)} \right)
$$

$$
\boldsymbol{w}^{(1)}, \boldsymbol{w}^{(2)}, ..., \boldsymbol{w}^{(n')} \sim \mu_t, \ \{X_i\}_{i=1}^{n} \sim \mathbb{P}_{\boldsymbol{\theta}}, \qquad \hat{G}^\dagger\mu_t(\boldsymbol{\theta}) \simeq \frac{1}{nn'} \sum_{i=1}^{n}\sum_{j=1}^{n'} f_{\boldsymbol{w}^{(j)}}\left( X_i \right).
$$

Now, assuming that we have obtained unbiased stochastic derivatives $-\sum_{s=1}^{t} \hat{G}^\dagger\mu_s$ and $\sum_{s=1}^{t}\left( -\hat{g} + \hat{G}\nu_s \right)$, how do we actually draw samples from $\mu_{t+1}$ and $\nu_{t+1}$? Provided we can answer this question, then we can start with two easy-to-sample distributions $(\mu_1, \nu_1)$, and then we will be able to draw samples from $(\mu_2, \nu_2)$. These samples in turn will allow us to draw samples from $(\mu_3, \nu_3)$, and so on. Therefore, it only remains to answer the above question. This leads us to:

**Step 3: Sampling by Stochastic Gradient Langevin Dynamics**

For any probability distribution with density function $e^{-h}\mathrm{d}\boldsymbol{z}$, the Stochastic Gradient Langevin Dynamics (SGLD) (Welling & Teh, 2011) iterates as

$$
\boldsymbol{z}_{k+1} = \boldsymbol{z}_k - \gamma\hat{\nabla}h(\boldsymbol{z}_k) + \sqrt{2\gamma}\epsilon\xi_k, \tag{8}
$$

where $\gamma$ is the step-size, $\hat{\nabla}h$ is an unbiased estimator of $\nabla h$, $\epsilon$ is the thermal noise, and $\xi_k \sim \mathcal{N}(0, I)$ is a standard normal vector, independently drawn across different iterations.

Suppose we start at $(\mu_1, \nu_1)$. Plugging $h \leftarrow -\hat{G}^\dagger\mu_1$ and $h \leftarrow -\hat{g} + \hat{G}\nu_1$ into (8), we obtain, for $\{X_i\}_{i=1}^{n} \sim \mathbb{P}_{\boldsymbol{\theta}_k}$, $\{\boldsymbol{w}^{(j)}\}_{j=1}^{n'} \sim \mu_1$, standard normal $\xi_k, \xi_k'$, and $X_i^{\mathrm{real}} \sim \mathbb{P}_{\mathrm{real}}$, $\{\boldsymbol{\theta}^{(j)}\}_{j=1}^{n'} \sim \nu_1$, $\{X_i^{(j)}\} \sim \mathbb{P}_{\boldsymbol{\theta}^{(j)}}$, the following update rules:

$$
\boldsymbol{\theta}_{k+1} = \boldsymbol{\theta}_k + \gamma\nabla_{\boldsymbol{\theta}} \left( \frac{1}{nn'} \sum_{i=1}^{n}\sum_{j=1}^{n'} f_{\boldsymbol{w}^{(j)}}\left( X_i \right) \right) + \sqrt{2\gamma}\epsilon\xi_k
$$

$$
\boldsymbol{w}_{k+1} = \boldsymbol{w}_k + \gamma\nabla_{\boldsymbol{w}} \left( \frac{1}{n} \sum_{i=1}^{n} f_{\boldsymbol{w}_k}(X_i^{\mathrm{real}}) - \frac{1}{nn'} \sum_{i=1}^{n}\sum_{j=1}^{n'} f_{\boldsymbol{w}_k}\left( X_i^{(j)} \right) \right) + \sqrt{2\gamma}\epsilon\xi_k'.
$$

The theory of (Welling & Teh, 2011) states that, for large enough $k$, the iterates of SGLD above (approximately) generate samples according to the probability measures $(\mu_2, \nu_2)$. We can then apply this process recursively to obtain samples from $(\mu_3, \nu_3), (\mu_4, \nu_4), ...(\mu_T, \nu_T)$. Finally, since the entropic MD and MP output the averaged measure $(\bar{\mu}_T, \bar{\nu}_T)$, it suffices to pick a random index $\hat{t} \in \{1, 2, ..., T\}$ and then output samples from $(\mu_{\hat{t}}, \nu_{\hat{t}})$.

Putting **Step 1-3** together, we obtain **Algorithm 4** and **5** in Appendix D.

*Remark.* In principle, any first-order sampling method is valid above. In the experimental section, we also use a RMSProp-preconditioned version of the SGLD (Li et al., 2016).

## 4.2 Summarizing Samples by Averaging: A Simple yet Effective Heuristic

Although **Algorithm 4** and **5** are implementable, they are quite complicated and resource-intensive, as the total computational complexity is $O(T^2)$. This high complexity comes from the fact that, when computing the stochastic derivatives, we need to store all the historical samples and evaluate new gradients at these samples.

An intuitive approach to alleviate the above issue is to try to summarize each distribution by only *one* parameter. To this end, the mean of the distribution is the most natural candidate, as it not only stablizes the algorithm, but also is often easier to acquire than the actual samples. For instance, computing the mean of distributions of the form $e^{-h}\mathrm{d}\boldsymbol{z}$, where $h$ is a loss function defined by deep neural networks, has been empirically proven successful in (Chaudhari et al., 2017; 2018; Dziugaite & Roy, 2018) via SGLD. In this paper, we adopt the same approach as in (Chaudhari et al., 2017) where we use exponential damping (the $\beta$ term in **Algorithm 3**) to increase stability. **Algorithm 3**, dubbed the *Mirror-GAN*, shows how to encompass this idea into entropic MD; the pseudocode for the similar *Mirror-Prox-GAN* can be found in **Algorithm 6** of Appendix D.

---

**Algorithm 3:** Mirror-GAN: Approximate Mirror Decent for GANs

**Input:** $\bar{\boldsymbol{w}}_1, \bar{\boldsymbol{\theta}}_1 \leftarrow$ random initialization, $\{\gamma_t\}_{t=1}^T, \{\epsilon_t\}_{t=1}^T, \{K_t\}_{t=1}^{T-1}, \beta$ (see Appendix D for meaning of the hyperparameters), standard normal noise $\xi_k, \xi_k'$.

**for** $t = 1, 2, \ldots, T-1$ **do**

    $\bar{\boldsymbol{w}}_t, \boldsymbol{w}_t^{(1)} \leftarrow \boldsymbol{w}_t$;

    $\bar{\boldsymbol{\theta}}_t, \boldsymbol{\theta}_t^{(1)} \leftarrow \boldsymbol{\theta}_t$;

    **for** $k = 1, 2, \ldots, K_t$ **do**

        Generate $A = \{X_1, \ldots, X_n\} \sim \mathbb{P}_{\boldsymbol{\theta}_t^{(k)}}$;

        $\boldsymbol{\theta}_t^{(k+1)} = \boldsymbol{\theta}_t^{(k)} + \frac{\gamma_t}{n} \nabla_{\boldsymbol{\theta}} \sum_{X_i \in A} f_{\boldsymbol{w}_t}(X_i) + \sqrt{2\gamma_t} \epsilon_t \xi_k$;

        Generate $B = \{X_1^{\mathrm{real}}, \ldots, X_n^{\mathrm{real}}\} \sim \mathbb{P}_{\mathrm{real}}$;

        Generate $B' = \{X_1', \ldots, X_n'\} \sim \mathbb{P}_{\boldsymbol{\theta}_t}$;

        $\boldsymbol{w}_t^{(k+1)} = \boldsymbol{w}_t^{(k)} + \frac{\gamma_t}{n} \nabla_{\boldsymbol{w}} \sum_{X_i^{\mathrm{real}} \in B} f_{\boldsymbol{w}_t^{(k)}}(X_i^{\mathrm{real}}) - \frac{\gamma_t}{n} \nabla_{\boldsymbol{w}} \sum_{X_i' \in B'} f_{\boldsymbol{w}_t^{(k)}}(X_i') + \sqrt{2\gamma_t} \epsilon_t \xi_k'$;

        $\bar{\boldsymbol{w}}_t \leftarrow (1-\beta)\bar{\boldsymbol{w}}_t + \beta \boldsymbol{w}_t^{(k+1)}$;

        $\bar{\boldsymbol{\theta}}_t \leftarrow (1-\beta)\bar{\boldsymbol{\theta}}_t + \beta \boldsymbol{\theta}_t^{(k+1)}$;

    $\boldsymbol{w}_{t+1} \leftarrow (1-\beta)\boldsymbol{w}_t + \beta \bar{\boldsymbol{w}}_t$;

    $\boldsymbol{\theta}_{t+1} \leftarrow (1-\beta)\boldsymbol{\theta}_t + \beta \bar{\boldsymbol{\theta}}_t$;

**return** $\boldsymbol{w}_T, \boldsymbol{\theta}_T$.

---

## 5 Experimental Evidence

The purpose of our experiments is twofold. First, we use established baselines to demonstrate that Mirror- and Mirror-Prox-GAN consistently achieve better or comparable performance than common algorithms. Second, we report that our algorithms are stable and always improve as the training process goes on. This is in contrast to unstable training algorithms, such as Adam, which often collapse to noise as the iteration count grows. (Cha, 2017).

We use visual quality of the generated images to evaluate different algorithms. We avoid reporting numerical metrics, as recent studies (Barratt & Sharma, 2018; Borji, 2018; Lucic et al., 2018) suggest that these metrics might be flawed. Setting of the hyperparameters and more auxiliary results can be found in Appendix E.

## 5.1 Synthetic Data

We repeat the synthetic setup as in (Gulrajani et al., 2017). The tasks include learning the distribution of 8 Gaussian mixtures, 25 Gaussian mixtures, and the Swiss Roll. For both the generator and discriminator, we use two MLPs with three hidden layers of 512 neurons. We choose SGD and Adam as baselines, and we compare them to Mirror- and Mirror-Prox-GAN. All algorithms are run up to $10^5$ iterations[4]. The results of 25 Gaussian mixtures are shown in Figure 1; An enlarged figure of 25 Gaussian Mixtures and other cases can be found in Appendix E.1.

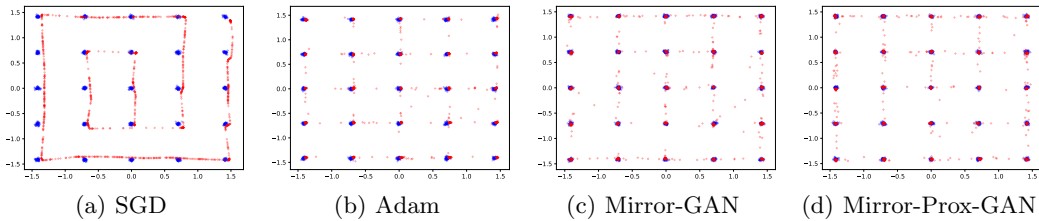

(a) SGD        (b) Adam        (c) Mirror-GAN        (d) Mirror-Prox-GAN

Figure 1: Fitting 25 Gaussian mixtures up to $10^5$ iterations. Blue dots represent the true distribution and red ones are from the trained generator.

As Figure 1 shows, SGD performs poorly in this task, while the other algorithms yield reasonable results. However, compared to Adam, Mirror- and Mirror-Prox-GAN fit the true distribution better in two aspects. First, the modes found by Mirror- and Mirror-Prox-GAN are more accurate than the ones by Adam, which are perceivably biased. Second, Mirror- and Mirror-Prox-GAN perform much better in capturing the variance (how spread the blue dots are), while Adam tends to collapse to modes. These observations are consistent throughout the synthetic experiments; see Appendix E.1.

## 5.2 Real Data

For real images, we use the `LSUN bedroom` dataset (Yu et al., 2015). We have also conducted a similar study with `MNIST`; more results can be found in Appendix E.2.

We use the same architecture (DCGAN) as in (Radford et al., 2015) with batch normalization. As the networks become deeper in this case, the gradient magnitudes differ significantly across different layers. As a result, non-adaptive methods such as SGD or SGLD do not perform well in this scenario. To alleviate such issues, we replace SGLD by the RMSProp-preconditioned SGLD (Li et al., 2016) for our sampling routines. For baselines, we consider two adaptive gradient methods: RMSprop and Adam. We also include two contemporary algorithms, the Simultaneous and Alternated Extra-Adam, from the concurrent ICLR submission (Gidel et al., 2018a).

Figure 2 shows the results at the $10^5$th iteration. The RMSProp, Alternated Extra-Adam and Mirror-GAN produce images with reasonable quality, while Adam and simultaneous Extra-Adam output noise. The visual quality of Alternated Extra-Adam and Mirror-GAN are comparable, and are better than RMSProp, as RMSProp sometimes generates blurry images (the $(3, 3)$- and $(1, 5)$-th entry of Figure 8.(b)).

It is worth mentioning that Adam can learn the true distribution at intermediate iterations, but later on suffers from mode collapse and finally degenerates to noise; see Appendix E.2.2.

---

[4]One iteration here means using one mini-batch of data. It does not correspond to the $T$ in our algorithms, as there might be multiple SGLD iterations within each time step $t$.

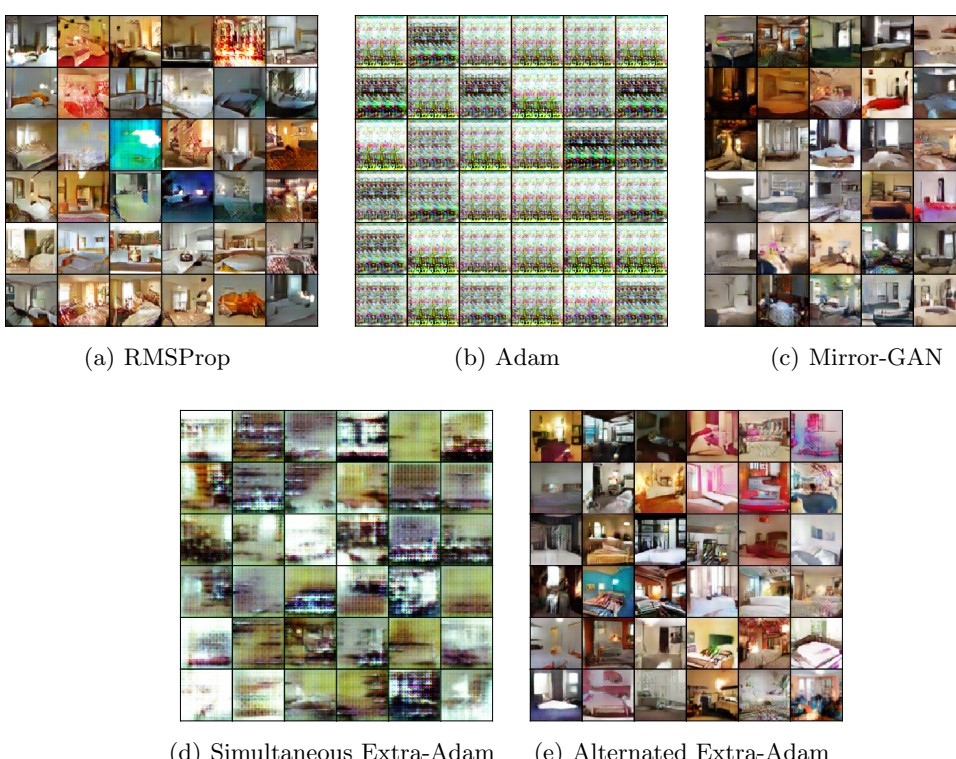

(a) RMSProp                    (b) Adam                    (c) Mirror-GAN

(d) Simultaneous Extra-Adam    (e) Alternated Extra-Adam

Figure 2: Dataset `LSUN bedroom`, $10^5$ iterations.

## 6    Conclusions

Our goal of systematically understanding and expanding on the game theoretic perspective of mixed NE along with stochastic Langevin dynamics for training GANs is a promising research vein. While simple in retrospect, we provide guidelines in developing approximate infinite-dimensional prox methods that mimic closely the provable optimization framework to learn the mixed NE of GANs. Our proposed Mirror- and Mirror-Prox-GAN algorithm feature cheap per-iteration complexity while rapidly converging to solutions of good quality.

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

# A    A Framework for Infinite-Dimensional Mirror Descent

## A.1    A note on the regularity

It is known that the (negative) Shannon entropy is *not* Fréchet differentiable in general. However, below we show that the Fréchet derive can be well-defined if we restrict the probability measures to within the set

$$\mathcal{M}(\mathcal{Z}) := \{\text{all probability measures on } \mathcal{Z} \text{ that admit densities w.r.t. the Lebesgue measure,}$$
$$\text{and the density is continuous and positive almost everywhere on } \mathcal{Z}\}.$$

We will also restrict the set of functions to be bounded and integrable:

$$\mathcal{F}(\mathcal{Z}) := \left\{\text{all bounded continuous functions } f \text{ on } \mathcal{Z} \text{ such that } \int e^{-f} < \infty\right\}.$$

It is important to notice that $\mu \in \mathcal{M}(\mathcal{Z})$ and $h \in \mathcal{F}(\mathcal{Z})$ implies $\mu' = \mathrm{MD}_\eta(\mu, h) \in \mathcal{M}(\mathcal{Z})$; this readily follows from the formula (7).

## A.2    Properties of Entropic Mirror Map

The total variation of a (possibly non-probability) measure $\mu \in \mathcal{M}(\mathcal{Z})$ is defined as (Halmos, 2013)

$$\|\mu\|_{\mathrm{TV}} = \sup_{\|h\|_{\mathbb{L}^\infty} \leq 1} \int h \mathrm{d}\mu = \sup_{\|h\|_{\mathbb{L}^\infty} \leq 1} \langle \mu, h \rangle.$$

Recall the standard topology induced by $\|\cdot\|_{\mathrm{TV}}$ and $\|\cdot\|_{\mathbb{L}^\infty}$ for measures and functions (Halmos, 2013), respectively. Whenever we speak about continuity or differentiability below, it is understood to be w.r.t. to the standard topology. Notice also that the $G$ operator defined in (5) is bounded if the discriminator $f_w$ is bounded, and hence continous (Halmos, 2013).

We depart from the fundamental *Gibbs Variational Principle*, which dates back to the earliest work of statistical mechanics (Gibbs, 1902). For two probability measures $\mu, \mu'$, denote their relative entropy by (the reason for this notation will become clear in (14))

$$D_\Phi(\mu, \mu') := \int_{\mathcal{Z}} \mathrm{d}\mu \log \frac{\mathrm{d}\mu}{\mathrm{d}\mu'}.$$

By the definition of $\mathcal{M}(\mathcal{Z})$, it is clear that the relative entropy is well-defined for any $\mu, \mu' \in \mathcal{M}(\mathcal{Z})$.

**Theorem 3** (*Gibbs Variation Principle*)**.** *Let $h \in \mathcal{F}(\mathcal{Z})$ and $\mu' \in \mathcal{M}(\mathcal{Z})$ be a reference measure. Then*

$$\log \int_{\mathcal{Z}} e^h \mathrm{d}\mu' = \sup_{\mu \in \mathcal{M}(\mathcal{Z})} \langle \mu, h \rangle - D_\Phi(\mu, \mu'), \tag{9}$$

*and equality is achieved by $\mathrm{d}\mu^\star = \frac{e^h \mathrm{d}\mu'}{\int_{\mathcal{Z}} e^h \mathrm{d}\mu'}$.*

Part of the following theorem is folklore in the mathematics and learning community. However, to the best of our knowledge, the relation to the entropic MD has not been systematically studied before, as we now do.

**Theorem 4.** *For a probability measure $\mathrm{d}\mu = \rho \mathrm{d}\boldsymbol{z}$, let $\Phi(\mu) = \int \rho \log \rho \mathrm{d}\boldsymbol{z}$ be the negative Shannon entropy, and let $\Phi^\star(h) = \log \int_{\mathcal{Z}} e^h \mathrm{d}\boldsymbol{z}$. Then*

*1. $\Phi^\star$ is the Fenchel conjugate of $\Phi$:*

$$\Phi^\star(h) = \sup_{\mu \in \mathcal{M}(\mathcal{Z})} \langle \mu, h \rangle - \Phi(\mu); \tag{10}$$

$$\Phi(\mu) = \sup_{h \in \mathcal{F}(\mathcal{Z})} \langle \mu, h \rangle - \Phi^\star(h). \tag{11}$$

2. *The derivatives admit the expression*

$$d\Phi(\mu) = 1 + \log \rho = \underset{h \in \mathcal{F}(\mathcal{Z})}{\arg\max} \langle \mu, h \rangle - \Phi^\star(h); \tag{12}$$

$$d\Phi^\star(h) = \frac{e^h d\mathbf{z}}{\int_{\mathcal{Z}} e^h d\mathbf{z}} = \underset{\mu \in \mathcal{M}(\mathcal{Z})}{\arg\max} \langle \mu, h \rangle - \Phi(\mu). \tag{13}$$

3. *The Bregman divergence of $\Phi$ is the relative entropy:*

$$D_\Phi(\mu, \mu') = \Phi(\mu) - \Phi(\mu') - \langle \mu - \mu', d\Phi(\mu') \rangle = \int_{\mathcal{Z}} d\mu \log \frac{d\mu}{d\mu'}. \tag{14}$$

4. *$\Phi$ is 4-strongly convex with respect to the total variation norm: For all $\lambda \in (0, 1)$,*

$$\Phi(\lambda\mu + (1-\lambda)\mu') \le \lambda\Phi(\mu) + (1-\lambda)\Phi(\mu') - \frac{1}{2} \cdot 4\lambda(1-\lambda)\|\mu - \mu'\|_{\mathrm{TV}}^2. \tag{15}$$

5. *The following duality relation holds: For any constant $C$, we have*

$$\forall \mu, \mu' \in \mathcal{M}(\mathcal{Z}), \quad D_\Phi(\mu, \mu') = D_{\Phi^\star}(d\Phi(\mu'), d\Phi(\mu)) = D_{\Phi^\star}(d\Phi(\mu') + C, d\Phi(\mu)). \tag{16}$$

6. *$\Phi^\star$ is $\frac{1}{4}$-smooth with respect to $\|\cdot\|_{\mathbb{L}^\infty}$:*

$$\forall h, h' \in \mathcal{F}(\mathcal{Z}), \quad \|d\Phi^\star(h) - d\Phi^\star(h')\|_{\mathrm{TV}} \le \frac{1}{4} \|h - h'\|_{\mathbb{L}^\infty}. \tag{17}$$

7. *Alternative to (17), we have the equivalent characterization of $\Phi^\star$:*

$$\forall h, h' \in \mathcal{F}(\mathcal{Z}), \quad \Phi^\star(h) \le \Phi^\star(h') + \langle d\Phi^\star(h'), h - h' \rangle + \frac{1}{2} \cdot \frac{1}{4} \|h - h'\|_{\mathbb{L}^\infty}^2. \tag{18}$$

8. *Similar to (16), we have*

$$\forall h, h', \quad D_{\Phi^\star}(h, h') = D_\Phi(d\Phi^\star(h'), d\Phi^\star(h)). \tag{19}$$

9. *The following three-point identity holds for all $\mu, \mu', \mu'' \in \mathcal{M}(\mathcal{Z})$:*

$$\langle \mu'' - \mu, d\Phi(\mu') - d\Phi(\mu) \rangle = D_\Phi(\mu, \mu') + D_\Phi(\mu'', \mu) - D_\Phi(\mu'', \mu'). \tag{20}$$

10. *Let the Mirror Descent iterate be defined as in (7). Then the following statements are equivalent:*

   (a) *$\mu_+ = \mathrm{MD}_\eta(\mu, h)$.*
   (b) *There exists a constant $C$ such that $d\Phi(\mu_+) = d\Phi(\mu) - \eta h + C$.*

   *In particular, for any $\mu', \mu'' \in \mathcal{M}(\mathcal{Z})$ we have*

$$\text{Let } \langle \mu' - \mu'', \eta h \rangle = \langle \mu' - \mu'', d\Phi(\mu) - d\Phi(\mu_+) \rangle. \tag{21}$$

*Proof.*

1. Equation (10) is simply the Gibbs variational principle (9) with $d\mu \leftarrow d\mathbf{z}$.

   By (10), we know that

$$\forall h \in \mathcal{F}(\mathcal{Z}), \quad \Phi(\mu) \ge \langle \mu, h \rangle - \log \int_{\mathcal{Z}} e^h d\mathbf{z}. \tag{22}$$

   But for $d\mu = \rho d\mathbf{z}$, the function $h := 1 + \log \rho$ saturates the equality in (22).

2. We prove a more general result on the Bregman divergence $D_\Phi$ in (23) below.

Let $\mathrm{d}\mu = \rho\mathrm{d}\mathbf{z}, \mathrm{d}\mu' = \rho'\mathrm{d}\mathbf{z}$, and $\mathrm{d}\mu'' = \rho''\mathrm{d}\mathbf{z} \in \mathcal{M}(\mathcal{Z})$. Let $\epsilon > 0$ be small enough such that $(\rho + \epsilon\rho'')\mathrm{d}\mathbf{z}$ is absolutely continuous with respect to $\mathrm{d}\mu'$; note that this is possible because $\mu, \mu'$, and $\mu'' \in \mathcal{M}(\mathcal{Z})$. We compute

$$
\begin{aligned}
D_\Phi(\rho + \epsilon\rho'', \rho') &= \int_{\mathcal{Z}} (\rho + \epsilon\rho'') \log \frac{\rho + \epsilon\rho''}{\rho'} \\
&= \int_{\mathcal{Z}} \rho \log \frac{\rho}{\rho'} + \int_{\mathcal{Z}} \rho \log\left(1 + \epsilon\frac{\rho''}{\rho}\right) + \epsilon\int_{\mathcal{Z}} \rho'' \log\frac{\rho}{\rho'} + \epsilon\int_{\mathcal{Z}} \rho'' \log\left(1 + \epsilon\frac{\rho''}{\rho}\right) \\
&\overset{(i)}{=} \int_{\mathcal{Z}} \rho \log\frac{\rho}{\rho'} + \epsilon\int_{\mathcal{Z}} \rho'' + \epsilon\int_{\mathcal{Z}} \rho'' \log\frac{\rho}{\rho'} + \epsilon^2\int_{\mathcal{Z}} \frac{\rho''^2}{\rho} + o(\epsilon) \\
&= D_\Phi(\rho, \rho') + \epsilon\int_{\mathcal{Z}} \rho''\left(1 + \log\frac{\rho}{\rho'}\right) + o(\epsilon),
\end{aligned}
$$

where (i) uses $\log(1 + t) = t + o(t)$ as $t \to 0$. In short, for all $\mu', \mu'' \in \mathcal{M}(\mathcal{Z})$,

$$
\mathrm{d}_\mu D_\Phi(\mu, \mu')(\mu'') = \left\langle \mu'', 1 + \log\frac{\rho}{\rho'} \right\rangle \tag{23}
$$

which means $\mathrm{d}_\mu D_\Phi(\mu, \mu') = 1 + \log\frac{\rho}{\rho'}$. The formula (12) is the special case when $\mathrm{d}\mu' \leftarrow \mathrm{d}\mathbf{z}$.

We now turn to (13). For every $h \in \mathcal{F}(\mathcal{Z})$, we need to show that the following holds for every $h' \in \mathcal{F}(\mathcal{Z})$:

$$
\Phi^\star(h + \epsilon h') - \Phi^\star(h) = \log\int_{\mathcal{Z}} e^{h + \epsilon h'}\mathrm{d}\mathbf{z} - \log\int_{\mathcal{Z}} e^h\mathrm{d}\mathbf{z} = \epsilon\int_{\mathcal{Z}} h'\frac{e^h}{\int_{\mathcal{Z}} e^h}\mathrm{d}\mathbf{z} + o(\epsilon). \tag{24}
$$

Define an auxiliary function

$$
T(\epsilon) := \log\int_{\mathcal{Z}} \frac{e^h}{\int_{\mathcal{Z}} e^h}e^{\epsilon h'}\mathrm{d}\mathbf{z}.
$$

Notice that $T(0) = 0$ and $T$ is smooth as a function of $\epsilon$. Thus, by the Intermediate Value Theorem,

$$
\begin{aligned}
\Phi^\star(h + \epsilon h') - \Phi^\star(h) &= T(\epsilon) - T(0) \\
&= (\epsilon - 0) \cdot \frac{\mathrm{d}}{\mathrm{d}\epsilon}T(\cdot)\Big|_{\epsilon'}
\end{aligned}
$$

for some $\epsilon' \in [0, \epsilon]$. A direct computation shows

$$
\frac{\mathrm{d}}{\mathrm{d}\epsilon}T(\cdot)\Big|_{\epsilon'} = \int_{\mathcal{Z}} h'\frac{e^{h + \epsilon'h'}}{\int_{\mathcal{Z}} e^{h + \epsilon'h'}}\mathrm{d}\mathbf{z}.
$$

Hence it suffices to prove $\frac{e^{h + \epsilon'h'}}{\int_{\mathcal{Z}} e^{h + \epsilon'h'}} = \frac{e^h}{\int_{\mathcal{Z}} e^h} + o(1)$ in $\epsilon$. To this end, let $C = \sup|h'| < \infty$. Then

$$
\frac{e^h}{\int_{\mathcal{Z}} e^h}e^{-2\epsilon'C} \le \frac{e^{h + \epsilon'h'}}{\int_{\mathcal{Z}} e^{h + \epsilon'h'}} \le \frac{e^h}{\int_{\mathcal{Z}} e^h}e^{2\epsilon'C}.
$$

It remains to use $e^t = 1 + t + o(t)$ and $\epsilon' \le \epsilon$.

3. Let $\mathrm{d}\mu = \rho\mathrm{d}\mathbf{z}$ and $\mathrm{d}\mu' = \rho'\mathrm{d}\mathbf{z}$. We compute

$$
\begin{aligned}
D_\Phi(\mu, \mu') &= \Phi(\mu) - \Phi(\mu') - \langle\mu - \mu', \mathrm{d}\Phi(\mu')\rangle \\
&= \int_{\mathcal{Z}} \rho \log \rho\mathrm{d}\mathbf{z} - \int_{\mathcal{Z}} \rho' \log \rho'\mathrm{d}\mathbf{z} - \langle\mu - \mu', 1 + \log \rho'\rangle \qquad \text{by (12)} \\
&= \int_{\mathcal{Z}} \rho \log\frac{\rho}{\rho'}\mathrm{d}\mathbf{z} \\
&= \int_{\mathcal{Z}} \mathrm{d}\mu \log\frac{\mathrm{d}\mu}{\mathrm{d}\mu'}.
\end{aligned}
$$

4. Define $\mu_\lambda = \lambda\mu + (1-\lambda)\mu'$. By (14) and the classical Pinsker's inequality (Gray, 2011), we have

$$\Phi(\mu) \geq \Phi(\mu_\lambda) + \langle (1-\lambda)(\mu-\mu'), \mathrm{d}\Phi(\mu_\lambda)\rangle + 2\|(1-\lambda)(\mu-\mu')\|_{\mathrm{TV}}^2, \qquad (25)$$

$$\Phi(\mu') \geq \Phi(\mu_\lambda) + \langle \lambda(\mu'-\mu), \mathrm{d}\Phi(\mu_\lambda)\rangle + 2\|\lambda(\mu-\mu')\|_{\mathrm{TV}}^2. \qquad (26)$$

Equation (15) follows by multiplying with $\lambda$ and $1-\lambda$ respectively and summing the two inequalities up.

5. Let $\mu = \rho\mathrm{d}\boldsymbol{z}$ and $\mu' = \rho'\mathrm{d}\boldsymbol{z}$. Then, by the definition of Bregman divergence and (12), (13),

$$D_{\Phi^\star}(\mathrm{d}\Phi(\mu'), \mathrm{d}\Phi(\mu)) = \Phi^\star(\mathrm{d}\Phi(\mu')) - \Phi^\star(\mathrm{d}\Phi(\mu)) - \left\langle \frac{e^{1+\log\rho}\mathrm{d}\boldsymbol{z}}{\int_{\mathcal{Z}} e^{1+\log\rho}}, 1 + \log\rho' - 1 - \log\rho \right\rangle$$

$$= \log\int_{\mathcal{Z}} e^{1+\log\rho'} - \log\int_{\mathcal{Z}} e^{1+\log\rho} + \int_{\mathcal{Z}} \rho\log\frac{\rho}{\rho'}$$

$$= \int_{\mathcal{Z}} \rho\log\frac{\rho}{\rho'} = D_\Phi(\mu, \mu')$$

since $\int_{\mathcal{Z}} \rho\mathrm{d}\boldsymbol{z} = \int_{\mathcal{Z}} \rho'\mathrm{d}\boldsymbol{z} = 1$. This proves the first equality.

For the second equality, we write

$$D_{\Phi^\star}(\mathrm{d}\Phi(\mu') + C, \mathrm{d}\Phi(\mu)) = \Phi^\star(\mathrm{d}\Phi(\mu') + C) - \Phi^\star(\mathrm{d}\Phi(\mu)) - \left\langle \frac{e^{1+\log\rho}\mathrm{d}\boldsymbol{z}}{\int_{\mathcal{Z}} e^{1+\log\rho}}, 1 + \log\rho' + C - 1 - \log\rho \right\rangle$$

$$= \log\int_{\mathcal{Z}} e^{1+\log\rho'+C} - \log\int_{\mathcal{Z}} e^{1+\log\rho} + \int_{\mathcal{Z}} \rho\log\frac{\rho}{\rho'} - C$$

$$= \int_{\mathcal{Z}} \rho\log\frac{\rho}{\rho'}$$

$$= D_\Phi(\mu, \mu') = D_{\Phi^\star}(\mathrm{d}\Phi(\mu'), \mathrm{d}\Phi(\mu))$$

where we have used the first equality in the last step.

6. Let $\mu_h = \mathrm{d}\Phi^\star(h)$, $\mu_{h'} = \mathrm{d}\Phi^\star(h')$, and $\mu_\lambda = \lambda\mu_h + (1-\lambda)\mu_{h'}$ for some $\lambda \in (0, 1)$. By Pinsker's inequality and (14), we have

$$\Phi(\mu_\lambda) \geq \Phi(\mu_h) + \langle \mu_\lambda - \mu_h, \mathrm{d}\Phi(\mu_h)\rangle + 2\|\mu_\lambda - \mu_h\|_{\mathrm{TV}}^2, \qquad (27)$$

$$\Phi(\mu_\lambda) \geq \Phi(\mu_{h'}) + \langle \mu_\lambda - \mu_{h'}, \mathrm{d}\Phi(\mu_{h'})\rangle + 2\|\mu_\lambda - \mu_{h'}\|_{\mathrm{TV}}^2. \qquad (28)$$

Now, notice that

$$\langle \mu_\lambda - \mu_h, \mathrm{d}\Phi(\mu_h)\rangle = \langle \mu_\lambda - \mu_h, \mathrm{d}\Phi(\mathrm{d}\Phi^\star(h))\rangle$$

$$= \left\langle \mu_\lambda - \mu_h, \mathrm{d}\Phi\left(\frac{e^h\mathrm{d}\boldsymbol{z}}{\int_{\mathcal{Z}} e^h}\right)\right\rangle \qquad \text{by (13)}$$

$$= \left\langle \mu_\lambda - \mu_h, 1 + h - \log\int_{\mathcal{Z}} e^h\right\rangle \qquad \text{by (12)}$$

$$= \langle \mu_\lambda - \mu_h, h\rangle$$

and, similarly, we have $\langle \mu_\lambda - \mu_{h'}, \mathrm{d}\Phi(\mu_{h'})\rangle = \langle \mu_\lambda - \mu_{h'}, h'\rangle$. Multiplying (27) by $\lambda$ and (28) by $1-\lambda$, summing the two up, and using the above equalities, we get

$$\Phi(\mu_\lambda) - \left(\lambda\Phi(\mu_h) + (1-\lambda)\Phi(\mu_{h'})\right) + \lambda(1-\lambda)\langle \mu_h - \mu_{h'}, h - h'\rangle \geq 2\lambda(1-\lambda)\|\mu_h - \mu_{h'}\|_{\mathrm{TV}}^2.$$

By (15), we know that

$$\Phi(\mu_\lambda) - \left(\lambda\Phi(\mu_h) + (1-\lambda)F(\mu_{h'})\right) \leq -2\lambda(1-\lambda)\|\mu_h - \mu_{h'}\|_{\mathrm{TV}}^2.$$

Moreover, by definition of the total variation norm, it is clear that

$$\langle \mu_h - \mu_{h'}, h - h'\rangle \leq \|\mu_h - \mu_{h'}\|_{\mathrm{TV}}\|h - h'\|_{\mathbb{L}^\infty}. \qquad (29)$$

Combing the last three inequalities gives (17).

7. Let $K$ be a positive integer and $k \in \{0, 1, 2, \ldots, K\}$. Set $\lambda_k = \frac{k}{K}$ and $h'' = h - h'$. Then

$$\Phi^\star(h) - \Phi^\star(h') = \Phi^\star(h' + \lambda_K h'') - \Phi^\star(h' + \lambda_0 h'')$$

$$= \sum_{k=0}^{K-1} \left( \Phi^\star(h' + \lambda_{k+1} h'') - \Phi^\star(h' + \lambda_k h'') \right). \tag{30}$$

By convexity of $\Phi^\star$, we have

$$\Phi^\star(h' + \lambda_{k+1} h'') - \Phi^\star(h' + \lambda_k h'') \leq \langle \mathrm{d}\Phi^\star(h' + \lambda_{k+1} h''), (\lambda_{k+1} - \lambda_k) h'' \rangle$$

$$= \frac{1}{K} \langle \mathrm{d}\Phi^\star(h' + \lambda_{k+1} h''), h'' \rangle. \tag{31}$$

By (29) and (17), we may further upper bound (31) as

$$\Phi^\star(h' + \lambda_{k+1} h'') - \Phi^\star(h' + \lambda_k h'') \leq \frac{1}{K} \left( \langle \mathrm{d}\Phi^\star(h'), h'' \rangle + \langle \mathrm{d}\Phi^\star(h' + \lambda_{k+1} h'') - \mathrm{d}\Phi^\star(h'), h'' \rangle \right)$$

$$\leq \frac{1}{K} \left( \langle \mathrm{d}\Phi^\star(h'), h'' \rangle + \| \mathrm{d}\Phi^\star(h' + \lambda_{k+1} h'') - \mathrm{d}\Phi^\star(h') \|_{\mathrm{TV}} \| h'' \|_{\mathbb{L}^\infty} \right)$$

$$\leq \frac{1}{K} \left( \langle \mathrm{d}\Phi^\star(h'), h'' \rangle + \frac{\lambda_{k+1}}{4} \| h'' \|_{\mathbb{L}^\infty}^2 \right). \tag{32}$$

Summing up (32) over $k$, we get, in view of (30),

$$\Phi^\star(h) - \Phi^\star(h') \leq \langle \mathrm{d}\Phi^\star(h'), h'' \rangle + \frac{1}{4} \| h'' \|_{\mathbb{L}^\infty}^2 \sum_{k=0}^{K-1} \lambda_{k+1}$$

$$= \langle \mathrm{d}\Phi^\star(h'), h'' \rangle + \frac{1}{4} \cdot \frac{K+1}{2K} \| h'' \|_{\mathbb{L}^\infty}^2. \tag{33}$$

Since $K$ is arbitrary, we may take $K \to \infty$ in (33), which is (18).

8. Straightforward calculation shows

$$D_{\Phi^\star}(h, h') = \log \int_{\mathcal{Z}} e^h - \log \int_{\mathcal{Z}} e^{h'} - \int_{\mathcal{Z}} \frac{e^{h'}}{\int e^{h'}} (h - h').$$

On the other hand, by definition of the Bregman divergence and (12), (13), we have

$$D_\Phi(\mathrm{d}\Phi^\star(h'), \mathrm{d}\Phi^\star(h)) = \int_{\mathcal{Z}} \frac{e^{h'}}{\int_{\mathcal{Z}} e^{h'}} h' - \log \int_{\mathcal{Z}} e^{h'} - \int_{\mathcal{Z}} \frac{e^h}{\int_{\mathcal{Z}} e^h} h + \log \int_{\mathcal{Z}} e^h$$

$$- \int_{\mathcal{Z}} \left( 1 + h - \log \int_{\mathcal{Z}} e^h \right) \left( \frac{e^{h'}}{\int_{\mathcal{Z}} e^{h'}} - \frac{e^h}{\int_{\mathcal{Z}} e^h} \right)$$

$$= \int_{\mathcal{Z}} \frac{e^{h'}}{\int e^{h'}} (h' - h) - \log \int_{\mathcal{Z}} e^{h'} + \log \int_{\mathcal{Z}} e^h$$

$$= \Phi^\star(h) - \Phi^\star(h') - \langle \mathrm{d}\Phi^\star(h'), h - h' \rangle$$

$$= D_{\Phi^\star}(h, h').$$

9. By definition of the Bregman divergence, we have

$$D_\Phi(\mu, \mu') = \Phi(\mu) - \Phi(\mu') - \langle \mu - \mu', \mathrm{d}\Phi(\mu') \rangle,$$
$$D_\Phi(\mu'', \mu) = \Phi(\mu'') - \Phi(\mu) - \langle \mu'' - \mu, \mathrm{d}\Phi(\mu) \rangle,$$
$$D_\Phi(\mu'', \mu') = \Phi(\mu'') - \Phi(\mu') - \langle \mu'' - \mu', \mathrm{d}\Phi(\mu') \rangle.$$

Equation (20) then follows by straightforward calculations.

10. First, let $\mu_+ = \mathrm{MD}_\eta(\mu, h)$. Then if $\mu_+ = \rho_+ \mathrm{d}\mathbf{z}$ and $\mu = \rho \mathrm{d}\mathbf{z}$, then (7) implies

$$\rho_+ = \frac{\rho e^{-\eta h}}{\int_{\mathcal{Z}} \rho e^{-\eta h}}.$$

By (12), we therefore have

$$d\Phi(\mu_+) = 1 + \log\rho_+$$

$$= 1 + \log\rho - \eta h - \log\int_{\mathcal{Z}}\rho e^{-\eta h}$$

whence (21) holds with $C = -\log\int_{\mathcal{Z}}\rho e^{-\eta h}$.

Conversely, assume that $d\Phi(\mu_+) = d\Phi(\mu) - \eta h + C$ for some constant $C$, and apply $d\Phi^\star$ to both sides. The left-hand side becomes

$$d\Phi^\star\Big(d\Phi(\mu_+)\Big) = d\Phi^\star(1 + \log\rho_+)$$

$$= \frac{\rho_+ d\boldsymbol{z}}{\int\rho_+ d\boldsymbol{z}} = \rho_+ d\boldsymbol{z} = d\mu_+,$$

where as the formula (13) implies that

$$d\Phi^\star\left(d\Phi(\mu) - \eta h + C\right) = \frac{e^{1+\log\rho-\eta h+C}}{\int_{\mathcal{Z}}e^{1+\log\rho-\eta h+C}}d\boldsymbol{z}$$

$$= \frac{\rho e^{-\eta h}d\boldsymbol{z}}{\int_{\mathcal{Z}}\rho e^{-\eta h}}$$

$$= \frac{e^{-\eta h}d\mu}{\int_{\mathcal{Z}}e^{-\eta h}d\mu}.$$

Combining the two equalities gives $d\mu_+ = \frac{e^{-\eta h}d\mu}{\int_{\mathcal{Z}}e^{-\eta h}d\mu}$ which exactly means $\mu_+ = \mathrm{MD}_\eta\left(\mu, h\right)$.

$\square$

# B  PROOF OF CONVERGENCE RATES FOR INFINITE-DIMENSIONAL MIRROR DESCENT

## B.1  MIRROR DESCENT, DETERMINISTIC DERIVATIVES

By the definition of the algorithm, (21), and the three-point identity (20), we have, for any $\mu \in \mathcal{M}(\mathcal{W})$,

$$\langle\mu_t - \mu, -g + G\nu_t\rangle = \frac{1}{\eta}\langle\mu_t - \mu, d\Phi(\mu_t) - d\Phi(\mu_{t+1})\rangle$$

$$= \frac{1}{\eta}\Big(D_\Phi(\mu, \mu_t) - D_\Phi(\mu, \mu_{t+1}) + D_\Phi(\mu_t, \mu_{t+1})\Big). \quad (34)$$

By item 10 of **Theorem 4**, there exists a constant $C_t$ such that

$$d\Phi(\mu_{t+1}) = d\Phi(\mu_t) - \eta\left(-g + G\nu_t\right) + C_t. \quad (35)$$

Using (16), we see that

$$D_\Phi(\mu_t, \mu_{t+1}) = D_{\Phi^\star}(d\Phi(\mu_{t+1}), d\Phi(\mu_t))$$

$$= D_{\Phi^\star}\Big(d\Phi(\mu_{t+1}) - C_t, d\Phi(\mu_t)\Big)$$

$$\leq \frac{1}{8}\|d\Phi(\mu_{t+1}) - C_t - d\Phi(\mu_t)\|_{\mathbb{L}^\infty}^2 \qquad \text{by (18)}$$

$$= \frac{\eta^2}{8}\|-g + G\nu_t\|_{\mathbb{L}^\infty}^2 \qquad \text{by (35)}$$

$$\leq \frac{\eta^2 M^2}{8}.$$

Consequently, we have

$$\sum_{t=1}^{T} \langle \mu_t - \mu, -g + G\nu_t \rangle = \sum_{t=1}^{T} \frac{1}{\eta} \Big( D_\Phi(\mu, \mu_t) - D_\Phi(\mu, \mu_{t+1}) + D_\Phi(\mu_t, \mu_{t+1}) \Big)$$

$$\leq \frac{D_\Phi(\mu, \mu_1)}{\eta} + \frac{\eta M^2 T}{8}. \tag{36}$$

Exactly the same argument applied to $\nu_t$'s yields, for any $\nu \in \mathcal{M}(\Theta)$,

$$\sum_{t=1}^{T} \langle \nu_t - \nu, -G^\dagger \mu_t \rangle \leq \frac{D_\Phi(\nu, \nu_1)}{\eta} + \frac{\eta M^2 T}{8}. \tag{37}$$

Summing up (36) and (37), substituting $\mu \leftarrow \mu_{\mathrm{NE}}, \nu \leftarrow \nu_{\mathrm{NE}}$ and dividing by $T$, we get

$$\frac{1}{T} \sum_{t=1}^{T} \Big( \langle \mu_t - \mu_{\mathrm{NE}}, -g + G\nu_t \rangle + \langle \nu_t - \nu_{\mathrm{NE}}, -G^\dagger \mu_t \rangle \Big) \leq \frac{D_0}{\eta T} + \frac{\eta M^2}{4}. \tag{38}$$

The left-hand side of (38) can be simplified to

$$\frac{1}{T} \sum_{t=1}^{T} \Big( \langle \mu_t - \mu_{\mathrm{NE}}, -g + G\nu_t \rangle + \langle \nu_t - \nu_{\mathrm{NE}}, -G^\dagger \mu_t \rangle \Big) = \frac{1}{T} \sum_{t=1}^{T} \Big( \langle \mu_{\mathrm{NE}} - \mu_t, g \rangle - \langle \mu_{\mathrm{NE}}, G\nu_t \rangle + \langle \mu_t, G\nu_{\mathrm{NE}} \rangle \Big)$$

$$= \langle \mu_{\mathrm{NE}}, g - G\bar{\nu}_T \rangle - \langle \bar{\mu}_T, g - G\nu_{\mathrm{NE}} \rangle. \tag{39}$$

By definition of the Nash Equilibrium, we have

$$\langle \bar{\mu}_T, g - G\nu_{\mathrm{NE}} \rangle \leq \langle \mu_{\mathrm{NE}}, g - G\nu_{\mathrm{NE}} \rangle \leq \langle \mu_{\mathrm{NE}}, g - G\bar{\nu}_T \rangle, \tag{40}$$
$$\langle \bar{\mu}_T, g - G\nu_{\mathrm{NE}} \rangle \leq \langle \bar{\mu}_T, g - G\bar{\nu}_T \rangle \leq \langle \mu_{\mathrm{NE}}, g - G\bar{\nu}_T \rangle,$$

which implies

$$|\langle \bar{\mu}_T, g - G\bar{\nu}_T \rangle - \langle \mu_{\mathrm{NE}}, g - G\nu_{\mathrm{NE}} \rangle| \leq \langle \mu_{\mathrm{NE}}, g - G\bar{\nu}_T \rangle - \langle \bar{\mu}_T, g - G\nu_{\mathrm{NE}} \rangle. \tag{41}$$

Combining (51)-(54), we conclude that

$$\eta = \frac{2}{M} \sqrt{\frac{D_0}{T}} \quad \Rightarrow \quad |\langle \bar{\mu}_T, g - G\bar{\nu}_T \rangle - \langle \mu_{\mathrm{NE}}, g - G\nu_{\mathrm{NE}} \rangle| \leq M \sqrt{\frac{D_0}{T}}.$$

## B.2 Mirror Descent, Stochastic Derivatives

We first write

$$\Big\langle \mu_t - \mu, \eta(-\hat{g} + \hat{G}\nu_t) \Big\rangle = \langle \mu_t - \mu, \eta(-g + G\nu_t) \rangle + \Big\langle \mu_t - \mu, \eta \big[ -\hat{g} + \hat{G}\nu_t + g - G\nu_t \big] \Big\rangle.$$

Taking conditional expectation and using the bias estimate of stochastic derivatives, we conclude that

$$\mathbb{E} \Big\langle \mu_t - \mu, \eta(-\hat{g} + \hat{G}\nu_t) \Big\rangle \leq \langle \mu_t - \mu, \eta(-g + G\nu_t) \rangle + \|\mu_t - \mu\|_{\mathrm{TV}} \cdot \eta\tau$$

$$\leq \langle \mu_t - \mu, \eta(-g + G\nu_t) \rangle + 2\eta\tau.$$

Therefore, using exactly the same argument leading to (36), we may obtain

$$\mathbb{E} \sum_{t=1}^{T} \Big\langle \mu_t - \mu, -\hat{g} + \hat{G}\nu_t \Big\rangle \leq \frac{\mathbb{E} D_\Phi(\mu, \mu_1)}{\eta} + \frac{\eta M'^2 T}{8} + 2\eta T\tau.$$

The rest is the same as with deterministic derivatives.

## C  Proof of Convergence Rates for Infinite-Dimensional Mirror-Prox

We first need a technical lemma, which is **Lemma 6.2** of (Juditsky & Nemirovski, 2011) tailored to our infinite-dimensional setting. We give a slightly different proof.

**Lemma 5.** *Given any* $\mu \in \mathcal{M}(\mathcal{Z})$ *and* $h, h' \in \mathcal{F}(\mathcal{Z})$, *let* $\mu = \mathrm{MD}_\eta(\tilde{\mu}, h)$ *and* $\tilde{\mu}_+ = \mathrm{MD}_\eta(\tilde{\mu}, h')$. *Let* $\Phi$ *be* $\alpha$-strongly convex (recall that $\alpha = 4$ when $\Phi$ is the entropy). *Then, for any* $\mu_\star \in \mathcal{M}(\mathcal{Z})$, *we have*

$$\langle \mu - \mu_\star, \eta h' \rangle \le D_\Phi(\mu_\star, \tilde{\mu}) - D_\Phi(\mu_\star, \tilde{\mu}_+) + \frac{\eta^2}{2\alpha} \|h - h'\|_{\mathbb{L}^\infty}^2 - \frac{\alpha}{2} \|\mu - \tilde{\mu}\|_{\mathrm{TV}}^2. \quad (42)$$

*Proof.* Recall from (15) that entropy is $\alpha$-strongly convex with respect to $\|\cdot\|_{\mathrm{TV}}$. We first write

$$\langle \mu - \mu_\star, \eta h' \rangle = \langle \tilde{\mu}_+ - \mu_\star, \eta h' \rangle + \langle \mu - \tilde{\mu}_+, \eta h \rangle + \langle \mu - \tilde{\mu}_+, \eta(h' - h) \rangle. \quad (43)$$

For the first term, (20) and (21) implies

$$\begin{aligned}
\langle \tilde{\mu}_+ - \mu_\star, \eta h' \rangle &= \langle \tilde{\mu}_+ - \mu_\star, \mathrm{d}\Phi(\tilde{\mu}) - \mathrm{d}\Phi(\tilde{\mu}_+) \rangle \\
&= -D_\Phi(\tilde{\mu}_+, \tilde{\mu}) - D_\Phi(\mu_\star, \tilde{\mu}_+) + D_\Phi(\mu_\star, \tilde{\mu}).
\end{aligned} \quad (44)$$

Similarly, the second term of the right-hand side of (43) can be written as

$$\langle \mu - \tilde{\mu}_+, \eta h \rangle = -D_\Phi(\mu, \tilde{\mu}) - D_\Phi(\tilde{\mu}_+, \mu) + D_\Phi(\tilde{\mu}_+, \tilde{\mu}). \quad (45)$$

Hölder's inequality for the third term gives

$$\begin{aligned}
\langle \mu - \tilde{\mu}_+, \eta(h' - h) \rangle &\le \|\mu - \tilde{\mu}_+\|_{\mathrm{TV}} \|\eta(h' - h)\|_{\mathbb{L}^\infty} \\
&\le \frac{\alpha}{2} \|\mu - \tilde{\mu}_+\|_{\mathrm{TV}}^2 + \frac{1}{2\alpha} \|\eta(h' - h)\|_{\mathbb{L}^\infty}^2.
\end{aligned} \quad (46)$$

Finally, recall that $\Phi$ is $\alpha$-strongly convex, and hence we have

$$-D_\Phi(\tilde{\mu}_+, \mu) \le -\frac{\alpha}{2} \|\mu - \tilde{\mu}_+\|_{\mathrm{TV}}^2, \quad -D_\Phi(\mu, \tilde{\mu}) \le -\frac{\alpha}{2} \|\mu - \tilde{\mu}\|_{\mathrm{TV}}^2. \quad (47)$$

The lemma follows by combining inequalities (44)-(47) in (43). $\qquad\square$

### C.1  Mirror-Prox, Deterministic Derivatives

Let $\alpha = 4$, $\bar{\mu}_T := \frac{1}{T} \sum_{t=1}^T \mu_t$, and $\bar{\nu}_T := \frac{1}{T} \sum_{t=1}^T \nu_t$.

In **Lemma 5**, substituting $\mu_\star \leftarrow \mu_{\mathrm{NE}}$, $\tilde{\mu} \leftarrow \tilde{\mu}_t$, $h \leftarrow -g + G\tilde{\nu}_t$ (so that $\mu = \mu_t$) and $h' \leftarrow -g + G\nu_t$ (so that $\tilde{\mu}_+ = \tilde{\mu}_{t+1}$), we get

$$\langle \mu_t - \mu_{\mathrm{NE}}, \eta(-g + G\nu_t) \rangle \le D_\Phi(\mu_{\mathrm{NE}}, \tilde{\mu}_t) - D_\Phi(\mu_{\mathrm{NE}}, \tilde{\mu}_{t+1}) + \frac{\eta^2}{2\alpha} \|G(\nu_t - \tilde{\nu}_t)\|_{\mathbb{L}^\infty}^2 - \frac{\alpha}{2} \|\tilde{\mu}_t - \mu_t\|_{\mathrm{TV}}^2. \quad (48)$$

Similarly, we have

$$\langle \nu_t - \nu_{\mathrm{NE}}, -\eta G^\dagger \mu_t \rangle \le D_\Phi(\nu_{\mathrm{NE}}, \tilde{\nu}_t) - D_\Phi(\nu_{\mathrm{NE}}, \tilde{\nu}_{t+1}) + \frac{\eta^2}{2\alpha} \left\|G^\dagger(\mu_t - \tilde{\mu}_t)\right\|_{\mathbb{L}^\infty}^2 - \frac{\alpha}{2} \|\tilde{\nu}_t - \nu_t\|_{\mathrm{TV}}^2. \quad (49)$$

Since $\|G(\nu_t - \tilde{\nu}_t)\|_{\mathbb{L}^\infty} \le L \cdot \|\nu_t - \tilde{\nu}_t\|_{\mathrm{TV}}$ and $\left\|G^\dagger(\mu_t - \tilde{\mu}_t)\right\|_{\mathbb{L}^\infty} \le L \cdot \|\mu_t - \tilde{\mu}_t\|_{\mathrm{TV}}$, summing up (48) and (49) yields

$$\begin{aligned}
\langle \mu_t - \mu_{\mathrm{NE}}, \eta(-g + G\nu_t) \rangle + \langle \nu_t - \nu_{\mathrm{NE}}, -\eta G^\dagger \mu_t \rangle &\le D_\Phi(\mu_{\mathrm{NE}}, \tilde{\mu}_t) - D_\Phi(\mu_{\mathrm{NE}}, \tilde{\mu}_{t+1}) + D_\Phi(\nu_{\mathrm{NE}}, \tilde{\nu}_t) - D_\Phi(\nu_{\mathrm{NE}}, \tilde{\nu}_{t+1}) \\
&\quad + \left(\frac{\eta^2 L^2}{2\alpha} - \frac{\alpha}{2}\right) \left(\|\tilde{\mu}_t - \mu_t\|_{\mathrm{TV}}^2 + \|\tilde{\nu}_t - \nu_t\|_{\mathrm{TV}}^2\right) \\
&\le D_\Phi(\mu_{\mathrm{NE}}, \tilde{\mu}_t) - D_\Phi(\mu_{\mathrm{NE}}, \tilde{\mu}_{t+1}) + D_\Phi(\nu_{\mathrm{NE}}, \tilde{\nu}_t) - D_\Phi(\nu_{\mathrm{NE}}, \tilde{\nu}_{t+1})
\end{aligned}$$

if $\eta \leq \frac{\alpha}{L} = \frac{4}{L}$. Summing up the last inequality over $t$ and using $D_\Phi(\cdot, \cdot) \geq 0$, we obtain

$$\frac{1}{T} \sum_{t=1}^{T} \left( \langle \mu_t - \mu_{\mathrm{NE}}, \eta(-g + G\nu_t) \rangle + \langle \nu_t - \nu_{\mathrm{NE}}, -\eta G^\dagger \mu_t \rangle \right) \leq \frac{D_\Phi(\mu_{\mathrm{NE}}, \tilde{\mu}_1) + D_\Phi(\nu_{\mathrm{NE}}, \tilde{\nu}_1)}{T} = \frac{D_0}{T}. \tag{50}$$

The left-hand side of (50) can be simplified to

$$\frac{1}{T} \sum_{t=1}^{T} \left( \langle \mu_t - \mu_{\mathrm{NE}}, \eta(-g + G\nu_t) \rangle + \langle \nu_t - \nu_{\mathrm{NE}}, -\eta G^\dagger \mu_t \rangle \right) = \frac{\eta}{T} \sum_{t=1}^{T} \left( \langle \mu_{\mathrm{NE}} - \mu_t, g \rangle - \langle \mu_{\mathrm{NE}}, G\nu_t \rangle + \langle \mu_t, G\nu_{\mathrm{NE}} \rangle \right)$$

$$= \eta \left( \langle \mu_{\mathrm{NE}}, g - G\bar{\nu}_T \rangle - \langle \bar{\mu}_T, g - G\nu_{\mathrm{NE}} \rangle \right). \tag{51}$$

By definition of the $(\mu_{\mathrm{NE}}, \nu_{\mathrm{NE}})$, we have

$$\begin{aligned} \langle \bar{\mu}_T, g - G\nu_{\mathrm{NE}} \rangle &\leq \langle \mu_{\mathrm{NE}}, g - G\nu_{\mathrm{NE}} \rangle \leq \langle \mu_{\mathrm{NE}}, g - G\bar{\nu}_T \rangle, \\ \langle \bar{\mu}_T, g - G\nu_{\mathrm{NE}} \rangle &\leq \;\; \langle \bar{\mu}_T, g - G\bar{\nu}_T \rangle \;\; \leq \langle \mu_{\mathrm{NE}}, g - G\bar{\nu}_T \rangle, \end{aligned} \tag{52}$$

which implies

$$|\langle \bar{\mu}_T, g - G\bar{\nu}_T \rangle - \langle \mu_{\mathrm{NE}}, g - G\nu_{\mathrm{NE}} \rangle| \leq \langle \mu_{\mathrm{NE}}, g - G\bar{\nu}_T \rangle - \langle \bar{\mu}_T, g - G\nu_{\mathrm{NE}} \rangle. \tag{53}$$

Combining (50)-(53), we conclude

$$\eta \leq \frac{4}{L} \quad \Rightarrow \quad |\langle \bar{\mu}_T, g - G\bar{\nu}_T \rangle - \langle \mu_{\mathrm{NE}}, g - G\nu_{\mathrm{NE}} \rangle| \leq \frac{D_0}{T\eta}.$$

### C.2 Mirror-Prox, Stochastic Derivatives

Let $\alpha = 4$, $\bar{\mu}_T := \frac{1}{T} \sum_{t=1}^{T} \mu_t$, and $\bar{\nu}_T := \frac{1}{T} \sum_{t=1}^{T} \nu_t$. Set the step-size to $\eta = \min\left[ \frac{\alpha}{\sqrt{3}L}, \sqrt{\frac{\alpha D_0}{6T\sigma^2}} \right]$.

In **Lemma 5**, substituting $\mu_\star \leftarrow \mu_{\mathrm{NE}}$, $\tilde{\mu} \leftarrow \tilde{\mu}_t$, $h \leftarrow -\hat{g} + \hat{G}\tilde{\nu}_t$ (so that $\mu = \mu_t$), and $h' \leftarrow -\hat{g} + \hat{G}\nu_t$ (so that $\tilde{\mu}_+ = \tilde{\mu}_{t+1}$), we get

$$\left\langle \mu_t - \mu_{\mathrm{NE}}, \eta(-\hat{g} + \hat{G}\nu_t) \right\rangle \leq D_\Phi(\mu_{\mathrm{NE}}, \tilde{\mu}_t) - D_\Phi(\mu_{\mathrm{NE}}, \tilde{\mu}_{t+1}) + \frac{\eta^2}{2\alpha} \left\| \hat{G}\nu_t - \hat{G}\tilde{\nu}_t \right\|_{\mathbb{L}^\infty}^2 - \frac{\alpha}{2} \|\tilde{\mu}_t - \mu_t\|_{\mathrm{TV}}^2. \tag{54}$$

Note that

$$\mathbb{E} \left\| \hat{G}\nu_t - \hat{G}\tilde{\nu}_t \right\|_{\mathbb{L}^\infty}^2 \leq 3 \left( \mathbb{E} \left\| \hat{G}\nu_t - G\nu_t \right\|_{\mathbb{L}^\infty}^2 + \mathbb{E} \|G\nu_t - G\tilde{\nu}_t\|_{\mathbb{L}^\infty}^2 + \mathbb{E} \left\| G\tilde{\nu}_t - \hat{G}\tilde{\nu}_t \right\|_{\mathbb{L}^\infty}^2 \right)$$

$$\leq 6\sigma^2 + 3L^2 \mathbb{E} \|\nu_t - \tilde{\nu}_t\|_{\mathrm{TV}}^2.$$

Therefore, taking expectation conditioned on the history for both sides of (54) and using the bias estimates of the stochastic derivatives, we get

$$\langle \mu_t - \mu_{\mathrm{NE}}, \eta(-g + G\nu_t) \rangle \leq \mathbb{E} D_\Phi(\mu_{\mathrm{NE}}, \tilde{\mu}_t) - \mathbb{E} D_\Phi(\mu_{\mathrm{NE}}, \tilde{\mu}_{t+1}) + \frac{3\eta^2 \sigma^2}{\alpha}$$

$$+ \frac{3\eta^2 L^2}{2\alpha} \mathbb{E} \|\nu_t - \tilde{\nu}_t\|_{\mathrm{TV}}^2 - \frac{\alpha}{2} \mathbb{E} \|\tilde{\mu}_t - \mu_t\|_{\mathrm{TV}}^2 + 2\eta\tau.$$

Similarly, we have

$$\langle \nu_t - \nu_{\mathrm{NE}}, -\eta G^\dagger \mu_t \rangle \leq \mathbb{E} D_\Phi(\nu_{\mathrm{NE}}, \tilde{\nu}_t) - \mathbb{E} D_\Phi(\nu_{\mathrm{NE}}, \tilde{\nu}_{t+1}) + \frac{3\eta^2 \sigma^2}{\alpha}$$

$$+ \frac{3\eta^2 L^2}{2\alpha} \mathbb{E} \|\mu_t - \tilde{\mu}_t\|_{\mathrm{TV}}^2 - \frac{\alpha}{2} \mathbb{E} \|\tilde{\nu}_t - \nu_t\|_{\mathrm{TV}}^2 + 2\eta\tau.$$

Summing up the last two inequalities over $t$ with $\eta \leq \frac{\alpha}{\sqrt{3}L}$ then yields

$$\frac{1}{T}\sum_{t=1}^{T}\Big(\langle\mu_t - \mu_{\mathrm{NE}}, -g + G\nu_t\rangle + \langle\nu_t - \nu_{\mathrm{NE}}, -G^{\dagger}\mu_t\rangle\Big)\Big) \leq \frac{D_0}{\eta T} + \frac{6\eta\sigma^2}{\alpha} + 4\tau$$

$$\leq \max\left[2\sqrt{\frac{6\sigma^2 D_0}{\alpha T}}, \frac{2\sqrt{3}LD_0}{\alpha T}\right] + 4\tau.$$

by definition of $\eta$. The rest is the same as with deterministic derivatives.

---

**Algorithm 4:** APPROX INF MIRROR DECENT

---

**Input:** $W[1], \Theta[1] \leftarrow n'$ samples from random initialization,
$\qquad \{\gamma_t\}_{t=1}^{T-1}, \{\epsilon_t\}_{t=1}^{T-1}, \{K\}_{t=1}^{T-1}, n, n'$, standard normal noise $\xi_k, \xi'_k$.

**for** $t = 1, 2, \ldots, T-1$ **do**

$\quad C \leftarrow \cup_{s=1}^{t} W[s], \quad D \leftarrow \cup_{s=1}^{t}\Theta[s]$ ;

$\quad \boldsymbol{w}_t^{(1)} \leftarrow \mathrm{UNIF}(W[t]), \quad \boldsymbol{\theta}_t^{(1)} \leftarrow \mathrm{UNIF}(\Theta[t])$;

$\quad$ **for** $k = 1, 2, \ldots, K_t, \ldots, K_t + n'$ **do**

$\quad\quad$ Generate $A = \{X_1, \ldots, X_n\} \sim \mathbb{P}_{\boldsymbol{\theta}_t^{(k)}}$;

$\quad\quad \boldsymbol{\theta}_t^{(k+1)} = \boldsymbol{\theta}_t^{(k)} + \frac{\gamma_t}{nn'}\nabla_{\boldsymbol{\theta}}\sum_{X_i \in A}\sum_{\boldsymbol{w} \in C} f_{\boldsymbol{w}}(X_i) + \sqrt{2\gamma_t}\epsilon_t\xi_k$;

$\quad\quad$ Generate $B = \{X_1^{\mathrm{real}}, \ldots, X_n^{\mathrm{real}}\} \sim \mathbb{P}_{\mathrm{real}}$;

$\quad\quad B' \leftarrow \{\}$ ;

$\quad\quad$ **for** each $\boldsymbol{\theta} \in D$ **do**

$\quad\quad\quad$ Generate $\tilde{B} = \{X_1', \ldots, X_n'\} \sim \mathbb{P}_{\boldsymbol{\theta}}$;

$\quad\quad\quad B' \leftarrow B' \cup \tilde{B}$;

$\quad\quad \boldsymbol{w}_t^{(k+1)} = \boldsymbol{w}_t^{(k)} + \frac{\gamma_t t}{n}\nabla_{\boldsymbol{w}}\sum_{X_i^{\mathrm{real}} \in B} f_{\boldsymbol{w}_t^{(k)}}(X_i^{\mathrm{real}}) - \frac{\gamma_t}{nn'}\nabla_{\boldsymbol{w}}\sum_{X_i' \in B'} f_{\boldsymbol{w}_t^{(k)}}(X_i') + \sqrt{2\gamma_t}\epsilon_t\xi'_k$;

$\quad W[t+1] \leftarrow \left\{\boldsymbol{w}_t^{(K+1)}, \ldots, \boldsymbol{w}_t^{(K+n')}\right\}, \quad \Theta[t+1] \leftarrow \left\{\boldsymbol{\theta}_t^{(K+1)}, \ldots, \boldsymbol{\theta}_t^{(K+n')}\right\}$;

$\mathtt{idx} \leftarrow \mathrm{UNIF}(1, 2, \ldots, T)$;

return $W[\mathtt{idx}], \Theta[\mathtt{idx}]$.

---

## D    OMITTED PSEUDOCODES IN THE MAIN TEXT

We use the following notation for the hyperparameters of our algorithms:

$$
\begin{aligned}
n &: \text{ number of samples in the data batch.} \\
n' &: \text{ number of samples for each probability measure.} \\
\gamma_t &: \text{ SGLD step-size at iteration } t. \\
\epsilon_t &: \text{ thermal noise of SGLD at iteration } t. \\
K_t &: \text{ warmup steps for SGLD at iteration } t. \\
\beta &: \text{ exponential damping factor in the weighted average.}
\end{aligned}
$$

The approximate infinite-dimensional entropic MD and MP in Section 4.1 are depicted in **Algorithm 4** and **5**, respectively. **Algorithm 6** gives the heuristic version of the entropic Mirror-Prox.

## E    DETAILS AND MORE RESULTS OF EXPERIMENTS

This section contains all the details regarding our experiments, as well as more results on synthetic and real datasets.

**Algorithm 5:** APPROX INF MIRROR-PROX

---

**Input:** $\tilde{W}[1], \tilde{\Theta}[1] \leftarrow n'$ samples from random initialization,
$\qquad \{\gamma_t\}_{t=1}^T, \{\epsilon_t\}_{t=1}^T, \{K_t\}_{t=1}^T, n, n'$, standard normal noise $\xi_k, \xi'_k, \xi''_k, \xi'''_k$.

**for** $t = 1, 2, \ldots, T$ **do**

$\quad C \leftarrow \tilde{W}[t] \cup \left(\cup_{s=1}^{t-1} W[s]\right), \quad D \leftarrow \tilde{\Theta}[t] \cup \left(\cup_{s=1}^{t-1} \Theta[s]\right)$ ;

$\quad \boldsymbol{w}_t^{(1)} \leftarrow \text{UNIF}(\tilde{W}[t]), \quad \boldsymbol{\theta}_t^{(1)} \leftarrow \text{UNIF}(\tilde{\Theta}[t])$;

$\quad$ **for** $k = 1, 2, \ldots, K_t, \ldots, K_t + n'$ **do**

$\qquad$ Generate $A = \{X_1, \ldots, X_n\} \sim \mathbb{P}_{\boldsymbol{\theta}_t^{(k)}}$;

$\qquad \boldsymbol{\theta}_t^{(k+1)} = \boldsymbol{\theta}_t^{(k)} + \frac{\gamma_t}{nn'} \nabla_{\boldsymbol{\theta}} \sum_{X_i \in A} \sum_{\boldsymbol{w} \in C} f_{\boldsymbol{w}}(X_i) + \sqrt{2\gamma_t}\epsilon_t \xi_k$;

$\qquad$ Generate $B = \{X_1^{\text{real}}, \ldots, X_n^{\text{real}}\} \sim \mathbb{P}_{\text{real}}$;

$\qquad B' \leftarrow \{\}$;

$\qquad$ **for** each $\boldsymbol{\theta} \in D$ **do**

$\qquad\quad$ Generate $\tilde{B} = \{X'_1, \ldots, X'_n\} \sim \mathbb{P}_{\boldsymbol{\theta}}$;

$\qquad\quad B' \leftarrow B' \cup \tilde{B}$;

$\qquad \boldsymbol{w}_t^{(k+1)} = \boldsymbol{w}_t^{(k)} + \frac{\gamma_t t}{n} \nabla_{\boldsymbol{w}} \sum_{X_i^{\text{real}} \in B} f_{\boldsymbol{w}_t^{(k)}}(X_i^{\text{real}}) - \frac{\gamma_t}{nn'} \nabla_{\boldsymbol{w}} \sum_{X'_i \in B'} f_{\boldsymbol{w}_t^{(k)}}(X'_i) + \sqrt{2\gamma_t}\epsilon_t \xi'_k$;

$\quad W[t] \leftarrow \left\{\boldsymbol{w}_t^{(K+1)}, \ldots, \boldsymbol{w}_t^{(K+n')}\right\}, \quad \Theta[t] \leftarrow \left\{\boldsymbol{\theta}_t^{(K+1)}, \ldots, \boldsymbol{\theta}_t^{(K+n')}\right\}$;

$\quad C' \leftarrow \cup_{s=1}^t W[s], \quad D' \leftarrow \cup_{s=1}^t \Theta[s]$ ;

$\quad \tilde{\boldsymbol{w}}_{t+1}^{(1)} \leftarrow \text{UNIF}(\tilde{W}[t]), \quad \tilde{\boldsymbol{\theta}}_{t+1}^{(1)} \leftarrow \text{UNIF}(\tilde{\Theta}[t])$ ;

$\quad$ **for** $k = 1, 2, \ldots, K_t, \ldots, K_t + n'$ **do**

$\qquad$ Generate $A = \{X_1, \ldots, X_n\} \sim \mathbb{P}_{\tilde{\boldsymbol{\theta}}_t^{(k)}}$;

$\qquad \tilde{\boldsymbol{\theta}}_{t+1}^{(k+1)} = \tilde{\boldsymbol{\theta}}_{t+1}^{(k)} + \frac{\gamma_t}{nn'} \nabla_{\boldsymbol{\theta}} \sum_{X_i \in A} \sum_{\boldsymbol{w} \in C'} f_{\boldsymbol{w}}(X_i) + \sqrt{2\gamma_t}\epsilon_t \xi''_k$;

$\qquad$ Generate $B = \{X_1^{\text{real}}, \ldots, X_n^{\text{real}}\} \sim \mathbb{P}_{\text{real}}$;

$\qquad B' \leftarrow \{\}$;

$\qquad$ **for** each $\boldsymbol{\theta} \in D'$ **do**

$\qquad\quad$ Generate $\tilde{B} = \{X'_1, \ldots, X'_n\} \sim \mathbb{P}_{\boldsymbol{\theta}}$;

$\qquad\quad B' \leftarrow B' \cup \tilde{B}$;

$\qquad \tilde{\boldsymbol{w}}_{t+1}^{(k+1)} = \tilde{\boldsymbol{w}}_{t+1}^{(k)} + \frac{\gamma_t t}{n} \nabla_{\boldsymbol{w}} \sum_{X_i^{\text{real}} \in B} f_{\tilde{\boldsymbol{w}}_{t+1}^{(k)}}(X_i^{\text{real}}) - \frac{\gamma_t}{nn'} \nabla_{\boldsymbol{w}} \sum_{X'_i \in B'} f_{\tilde{\boldsymbol{w}}_{t+1}^{(k)}}(X'_i) + \sqrt{2\gamma_t}\epsilon_t \xi'''_k$);

$\quad \tilde{W}[t+1] \leftarrow \left\{\tilde{\boldsymbol{w}}_{t+1}^{(K+1)}, \ldots, \tilde{\boldsymbol{w}}_{t+1}^{(K+n')}\right\}, \quad \tilde{\Theta}[t+1] \leftarrow \left\{\tilde{\boldsymbol{\theta}}_{t+1}^{(K+1)}, \ldots, \tilde{\boldsymbol{\theta}}_{t+1}^{(K+n')}\right\}$;

$\texttt{idx} \leftarrow \text{UNIF}(1, 2, \ldots, T)$;

return $W[\texttt{idx}], \Theta[\texttt{idx}]$.

---

**Network Architectures:** For all experiments, we consider the gradient-penalized discriminator (Gulrajani et al., 2017) as a soft constraint alternative to the original Wasserstein GANs, as it is known to achieve much better performance. The gradient penalty parameter is denoted by $\lambda$ below.

For synthetic data, we use fully connected networks for both the generator and discriminator. They consist of three layers, each of them containing 512 neurons, with ReLU as nonlinearity.

For `MNIST`, we use convolutional neural networks identical to (Gulrajani et al., 2017) as the generator and discriminator.[5] The generator uses a sigmoid function to map the output to range $[0, 1]$.

---

[5] Their code is available on `https://github.com/igul222/improved_wgan_training`.

---

**Algorithm 6:** MIRROR-PROX-GAN: APPROXIMATE MIRROR-PROX FOR GANS

---

**Input:** $\tilde{\boldsymbol{w}}_1, \tilde{\boldsymbol{\theta}}_1 \leftarrow$ random initialization,
$\qquad \boldsymbol{w}_0 \leftarrow \tilde{\boldsymbol{w}}_1, \boldsymbol{\theta}_0 \leftarrow \tilde{\boldsymbol{\theta}}_1, \{\gamma_t\}_{t=1}^T, \{\epsilon_t\}_{t=1}^T, \{K_t\}_{t=1}^T, \beta,$ standard normal noise
$\qquad \xi_k, \xi_k', \xi_k'', \xi_k'''.$
**for** $t = 1, 2, \dots, T$ **do**
$\quad \bar{\boldsymbol{w}}_t, \bar{\boldsymbol{w}}_{t+1}, \tilde{\boldsymbol{w}}_t^{(1)}, \tilde{\boldsymbol{w}}_{t+1}^{(1)} \leftarrow \tilde{\boldsymbol{w}}_t, \quad \bar{\boldsymbol{\theta}}_t, \bar{\boldsymbol{\theta}}_{t+1}, \tilde{\boldsymbol{\theta}}_t^{(1)}, \tilde{\boldsymbol{\theta}}_{t+1}^{(1)} \leftarrow \tilde{\boldsymbol{\theta}}_t;$
$\quad$ **for** $k = 1, 2, \dots, K_t$ **do**
$\qquad$ Generate $A = \{X_1, \dots, X_n\} \sim \mathbb{P}_{\boldsymbol{\theta}_t^{(k)}};$
$\qquad \boldsymbol{\theta}_t^{(k+1)} = \boldsymbol{\theta}_t^{(k)} + \frac{\gamma_t}{n} \nabla_{\boldsymbol{\theta}} \sum_{X_i \in A} f_{\tilde{\boldsymbol{w}}_t}(X_i) + \sqrt{2\gamma_t} \epsilon_t \xi_k;$
$\qquad$ Generate $B = \{X_1^{\text{real}}, \dots, X_n^{\text{real}}\} \sim \mathbb{P}_{\text{real}};$
$\qquad$ Generate $B' = \{X_1', \dots, X_n'\} \sim \mathbb{P}_{\tilde{\boldsymbol{\theta}}_t};$

$\qquad \boldsymbol{w}_t^{(k+1)} = \boldsymbol{w}_t^{(k)} + \frac{\gamma_t}{n} \nabla_{\boldsymbol{w}} \sum_{X_i^{\text{real}} \in B} f_{\boldsymbol{w}_t^{(k)}}(X_i^{\text{real}}) - \frac{\gamma_t}{n} \nabla_{\boldsymbol{w}} \sum_{X_i' \in B'} f_{\boldsymbol{w}_t^{(k)}}(X_i') + \sqrt{2\gamma_t} \epsilon_t \xi_k';$

$\qquad \bar{\boldsymbol{w}}_t \leftarrow (1 - \beta) \bar{\boldsymbol{w}}_t + \beta \boldsymbol{w}_t^{(k+1)};$
$\qquad \bar{\boldsymbol{\theta}}_t \leftarrow (1 - \beta) \bar{\boldsymbol{\theta}}_t + \beta \boldsymbol{\theta}_t^{(k+1)} \;;$
$\quad \boldsymbol{w}_t \leftarrow (1 - \beta) \boldsymbol{w}_{t-1} + \beta \bar{\boldsymbol{w}}_t;$
$\quad \boldsymbol{\theta}_t \leftarrow (1 - \beta) \boldsymbol{\theta}_{t-1} + \beta \bar{\boldsymbol{\theta}}_t;$

$\quad$ **for** $k = 1, 2, \dots, K_t$ **do**
$\qquad$ Generate $A = \{X_1, \dots, X_n\} \sim \mathbb{P}_{\tilde{\boldsymbol{\theta}}_{t+1}^{(k)}};$
$\qquad \tilde{\boldsymbol{\theta}}_{t+1}^{(k+1)} = \tilde{\boldsymbol{\theta}}_{t+1}^{(k)} + \frac{\gamma_t}{n} \nabla_{\boldsymbol{\theta}} \sum_{X_i \in A} f_{\boldsymbol{w}_t}(X_i) + \sqrt{2\gamma_t} \epsilon_t \xi_k'';$
$\qquad$ Generate $B = \{X_1^{\text{real}}, \dots, X_n^{\text{real}}\} \sim \mathbb{P}_{\text{real}};$
$\qquad$ Generate $B' = \{X_1', \dots, X_n'\} \sim \mathbb{P}_{\boldsymbol{\theta}_t};$

$\qquad \boldsymbol{w}_{t+1}^{(k+1)} = \boldsymbol{w}_{t+1}^{(k)} + \frac{\gamma_t}{n} \nabla_{\boldsymbol{w}} \sum_{X_i^{\text{real}} \in B} f_{\boldsymbol{w}_{t+1}^{(k)}}(X_i^{\text{real}}) - \frac{\gamma_t}{n} \nabla_{\boldsymbol{w}} \sum_{X_i' \in B'} f_{\boldsymbol{w}_{t+1}^{(k)}}(X_i') + \sqrt{2\gamma_t} \epsilon_t \xi_k''';$

$\qquad \bar{\boldsymbol{w}}_{t+1} \leftarrow (1 - \beta) \bar{\boldsymbol{w}}_{t+1} + \beta \boldsymbol{w}_{t+1}^{(k+1)};$
$\qquad \bar{\boldsymbol{\theta}}_{t+1} \leftarrow (1 - \beta) \bar{\boldsymbol{\theta}}_{t+1} + \beta \boldsymbol{\theta}_{t+1}^{(k+1)} \;;$
$\quad \tilde{\boldsymbol{w}}_{t+1} \leftarrow (1 - \beta) \tilde{\boldsymbol{w}}_t + \beta \bar{\boldsymbol{w}}_{t+1};$
$\quad \tilde{\boldsymbol{\theta}}_{t+1} \leftarrow (1 - \beta) \tilde{\boldsymbol{\theta}}_t + \beta \bar{\boldsymbol{\theta}}_{t+1};$
**return** $\boldsymbol{w}_T, \boldsymbol{\theta}_T.$

---

For `LSUN bedroom`, we use DCGAN (Radford et al., 2015), except that the number of the channels in each layer is half of the original model, and the last sigmoid function of the discriminator is removed. The output of the generator is mapped to $[0, 1]$ by hyperbolic tangent and a linear transformation. The architecture contains batch normalization layer to ensure the stability of the training. For our Mirror- and Mirror-Prox-GAN, the Gaussian noise from SGLD is not added to parameters in batch normalization layers, as the batch normalization creates strong dependence among entries of the weight matrix and was not covered by our theory.

**Hyperparameter setting:** The hyperparameter setting is summarized in Table 1. For baselines (SGD, RMSProp, Adam), we use the settings identical to (Gulrajani et al., 2017). For our proposed Mirror- and Mirror-Prox-GAN, we set the damping factor $\beta$ to be 0.9. For

| Algorithm | SGD | | RMSProp | Adam | | | Entropic MD/MP | | |
|---|---|---|---|---|---|---|---|---|---|
| Dataset | S | M | L | S | M | L | S | M | L |
| Step-size $\gamma$ | $10^{-2}$ | | $10^{-4}$ | $10^{-4}$ | | | $10^{-2}$ | | $10^{-4}$ |
| Gradient penalty $\lambda$ | 0.1 | | 10 | 0.1 | 10 | | 0.1 | | 10 |
| Noise $\epsilon$ | | | | | | | $10^{-2}$ | $10^{-3}$ | $10^{-6}$ |
| Batch Size $n$ | 1024 | 50 | 64 | 1024 | 50 | 64 | 1024 | 50 | 64 |

Table 1: Hyperparameter setting. "S", "M", "L" stands for synthetic data, `MNIST` and `LSUN bedroom`, respectively. MD for `LSUN bedroom` uses a RMSProp preconditioner, so the step-size is the same as one in RMSProp.

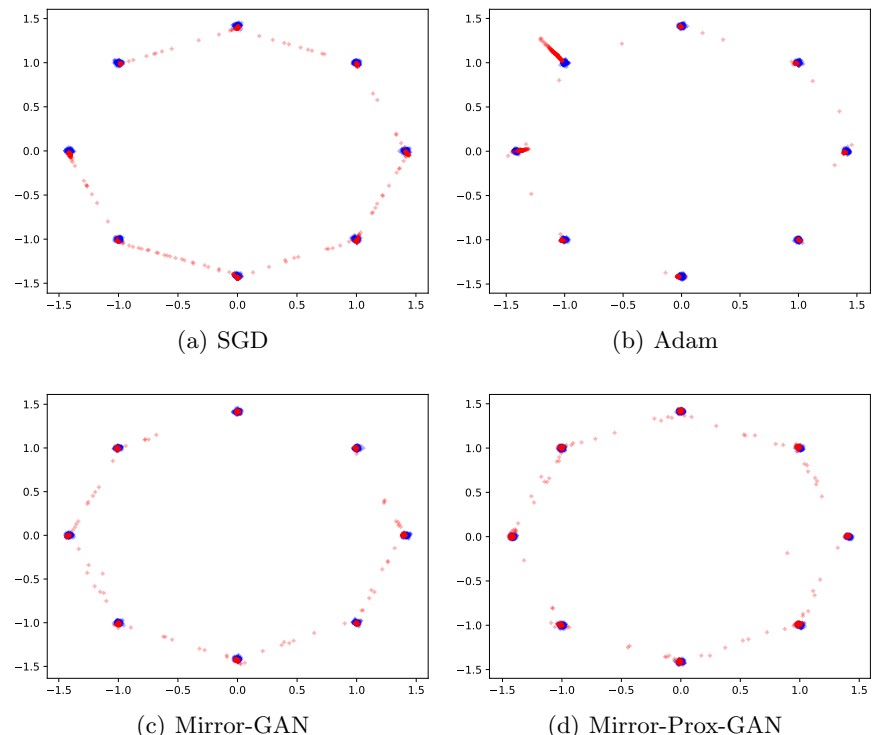

(a) SGD  (b) Adam

(c) Mirror-GAN  (d) Mirror-Prox-GAN

Figure 3: Fitting 8 Gaussian mixtures up to $10^5$ iterations.

$K_t, \gamma_t$ and $\epsilon_t$, we use the simple exponential scheduling:

$$K_t = \lfloor (1 + 10^{-5})^t \rfloor.$$
$$\gamma_t = \gamma \times (1 - 10^{-5})^t, \qquad \gamma \text{ in Table 1.}$$
$$\epsilon_t = \epsilon \times (1 - 5 \times 10^{-5})^t, \quad \epsilon \text{ in Table 1.}$$

The idea is that the initial iterations are very noisy, and hence it makes sense to take less SGLD steps. As the iteration counts grow, the algorithms learn more meaningful parameters, and we should increase the number of SGLD steps as well as decreasing the step-size $\gamma_t$ and thermal noise $\epsilon_t$ to make the sampling more accurate. This is akin to the warmup steps in the sampling literature.

### E.1 Synthetic Data

Figure 3, 4, and 5 show results on learning 8 Gaussian mixtures, 25 Gaussian mixtures, and the Swiss Roll. As in the case for 25 Gaussian mixtures, we find that Mirror- and Mirror-

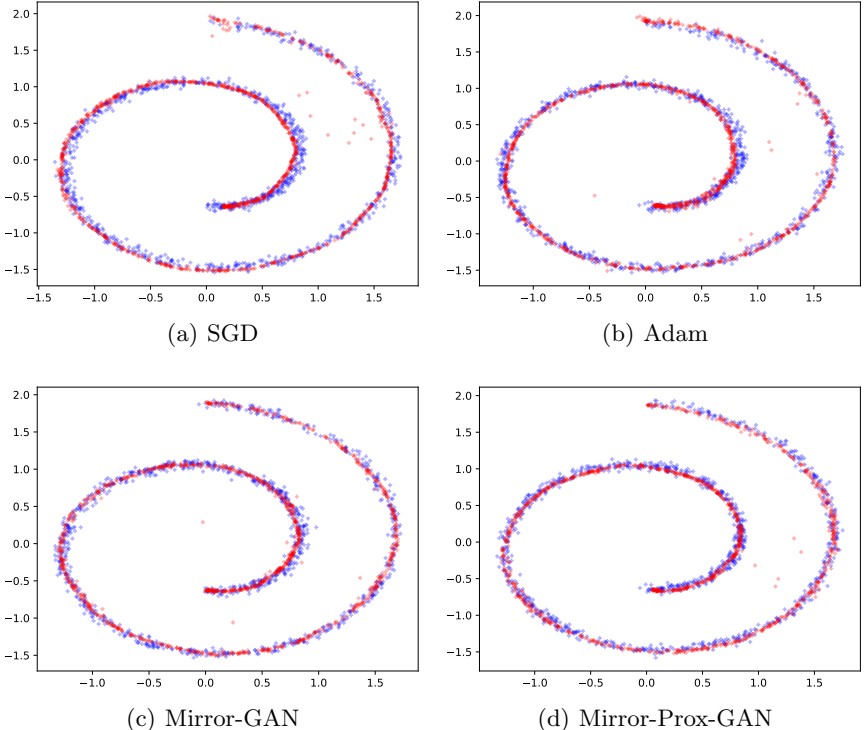

Figure 4: Fitting the 'Swiss Roll' up to $10^5$ iterations.

Prox-GAN can better capture the variance of the true distribution, as well as finding the unbiased modes.

In Figure 6, we plot the data generated after $10^4, 2 \times 10^4, 5 \times 10^4, 8 \times 10^4$, and $10^5$ iterations by different algorithms fro 25 Gaussian mixtures. It is clear that Mirror- and Mirror-Prox-GAN find the modes of the distribution faster. In practice, it was observed that the noise introduced by SGLD quickly drives the iterates to non-trivial parameter regions, whereas SGD tends to get stuck at very bad local minima. Adam, as an adaptive algorithm, is capable of escaping bad local minima, however at a rate slower than Mirror- and Mirror-Prox-GAN. The quality of Adam's final solution is also not as good as Mirror- and Mirror-Prox-GAN; see the discussions in Section 5.1.

## E.2 REAL DATA

### E.2.1 MNSIT

Results on `MNIST` dataset are shown in Figure 7. The models are trained by each algorithm for $10^5$ iterations. We can see that all algorithms achieve comparable performance. Therefore, the dataset seems too weak to be a discriminator for different algorithms.

### E.2.2 LSUN BEDROOM

| Algorithm | RMSProp | Adam | Entropic MD | Extra-Adam |
|---|---|---|---|---|
| Simultaneous | - | - | 3.0955 | 2.0015 |
| Alternated | 3.0555 | 1.3730 | - | 3.1620 |

Table 2: Inception Score of generator trained on LSUN dataset. The reported scores are based on the average of 6400 images from each generator.

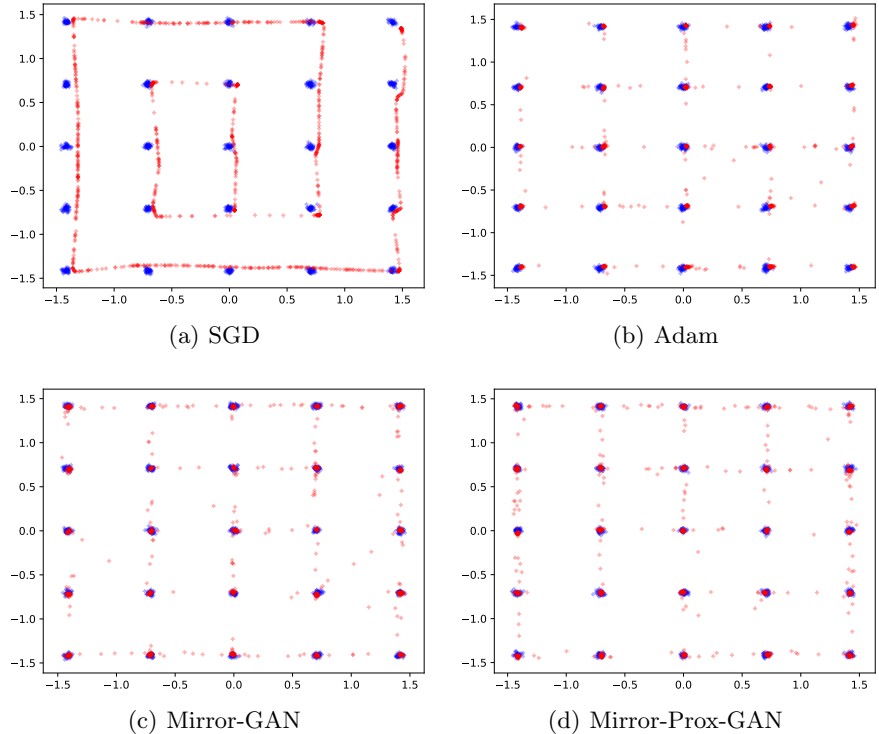

Figure 5: Fitting 25 Gaussian mixtures up to $10^5$ iterations.

More results on the `LSUN bedroom` dataset are shown in Figure 8. We show images generated after $4 \times 10^4, 8 \times 10^4$, and $10^5$ iterations by each algorithm. We can see that the Mirror-GAN and Alternated Extra-Adam outperform vanilla RMSProp. Adam was able to obtain meaningful images in early stages of training. However, further iterations do not improve the image quality of Adam. In contrast, they lead to severe mode collapse at the $8 \times 10^4$th iteration, and converge to noise later on. Simultaneous Extra-Adam completely fails in this task.

Finally, for reference, we report the Inception Score in Table 2.

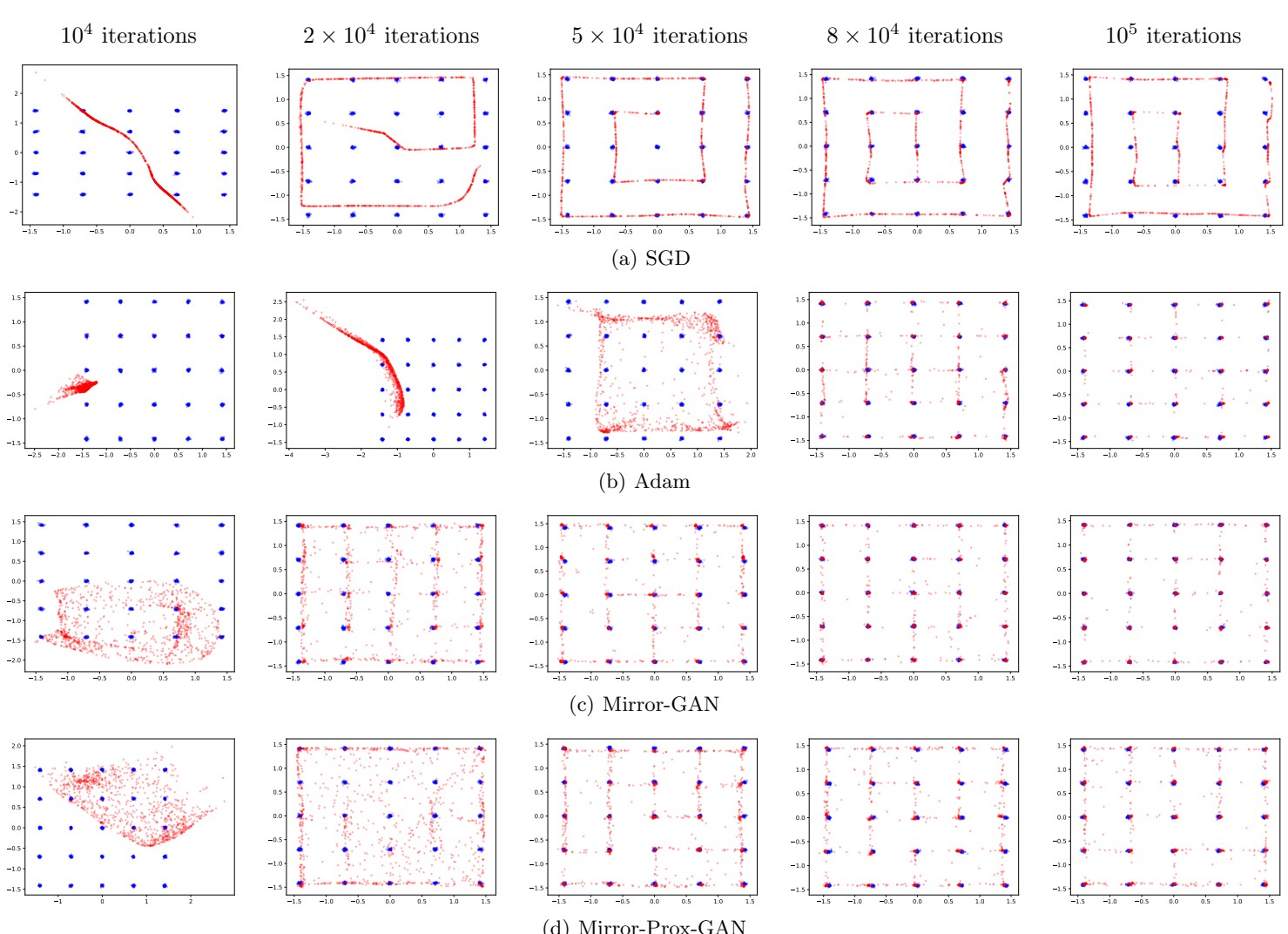

Figure 6: Learning 25 Gaussian mixtures accross different iterations.

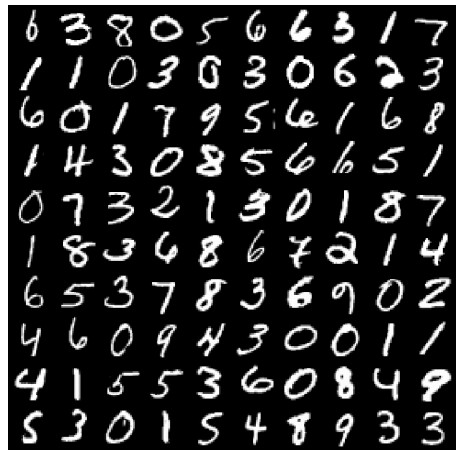

(a) True Data

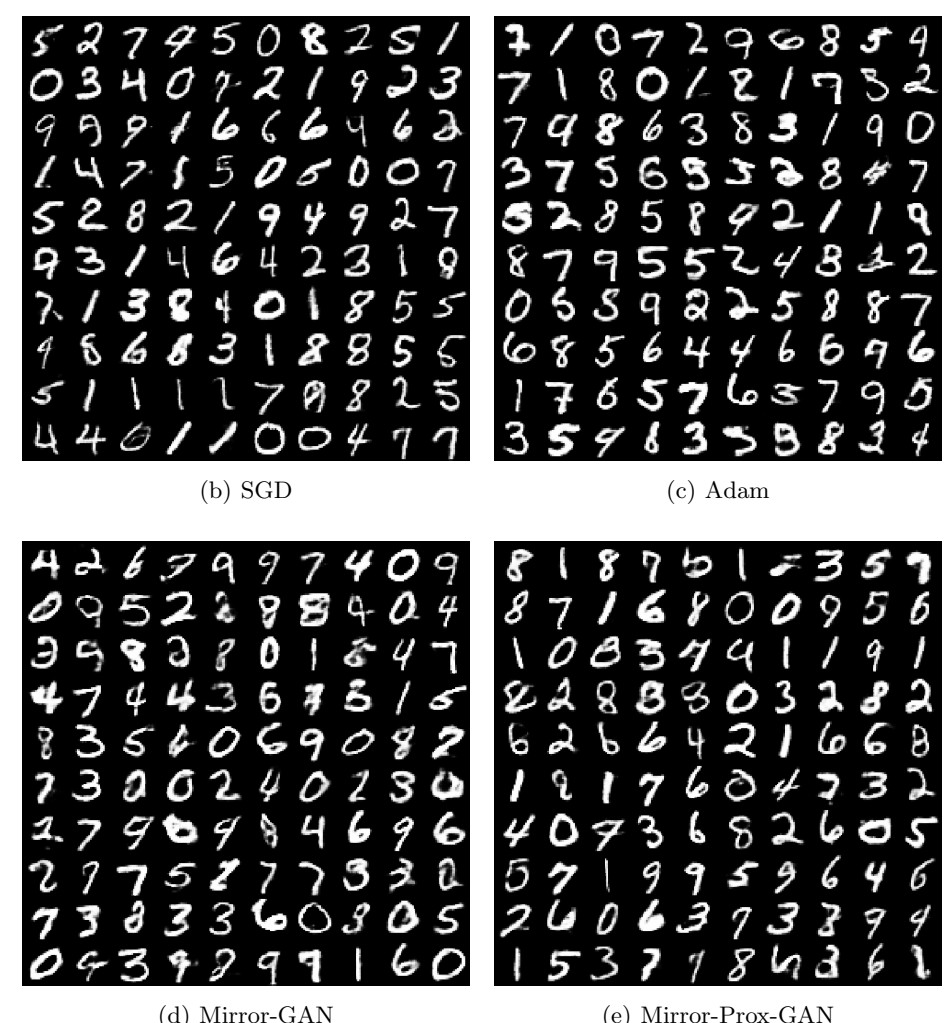

(b) SGD            (c) Adam

(d) Mirror-GAN          (e) Mirror-Prox-GAN

Figure 7: True `MNIST` images and samples generated by different algorithms.

$4 \times 10^4$ iterations $\qquad$ $8 \times 10^4$ iterations $\qquad$ $10^5$ iterations

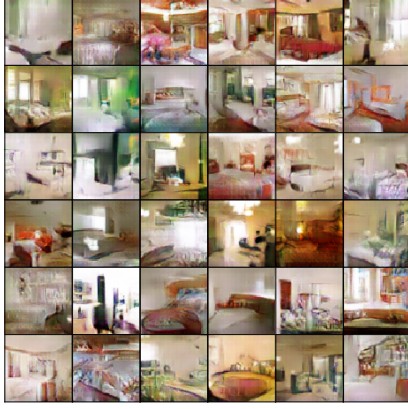 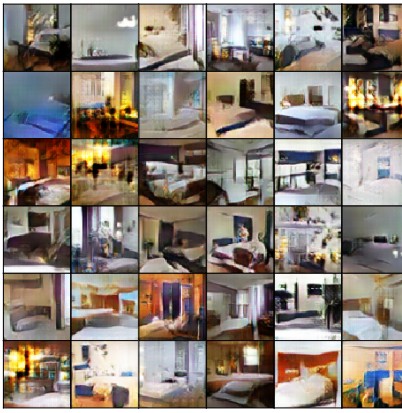 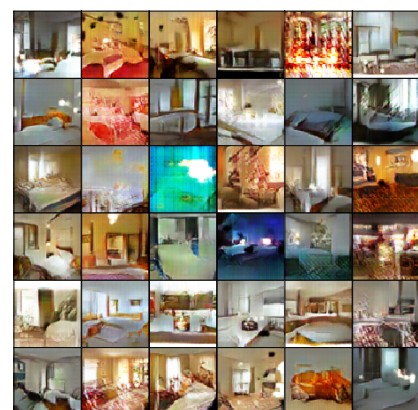

(a) RMSProp

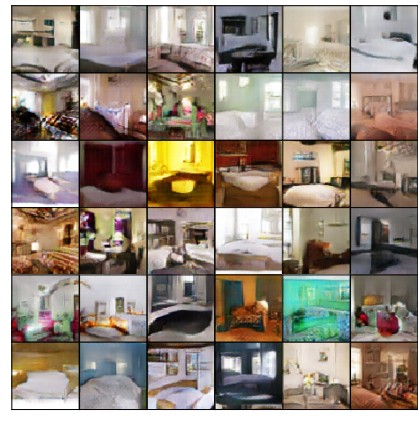 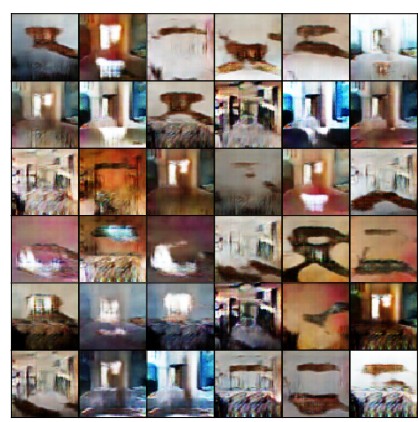

(b) Adam

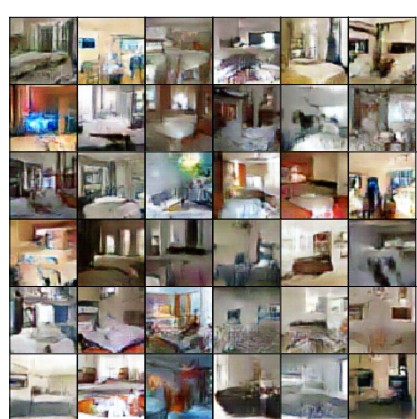 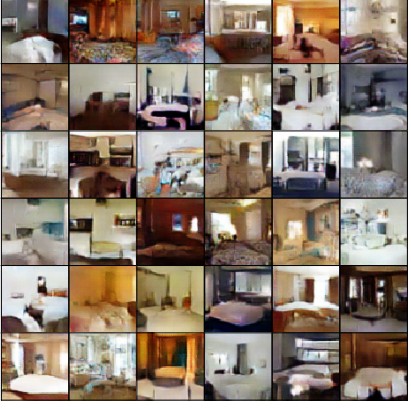

(c) Mirror-GAN, **Algorithm 3**

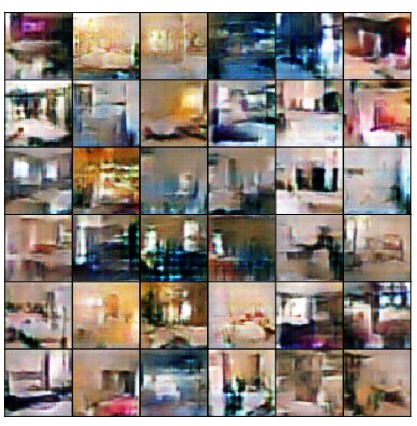 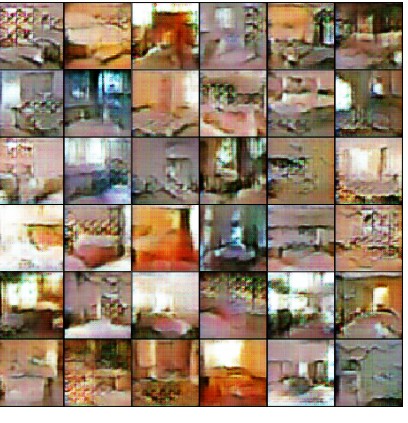 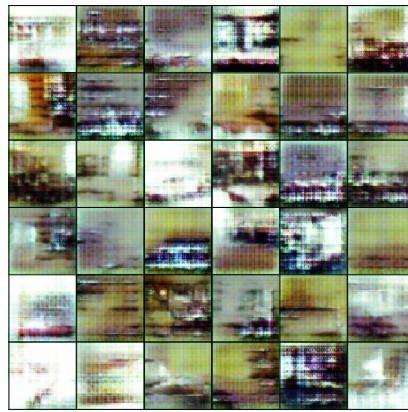

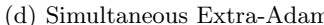

(d) Simultaneous Extra-Adam

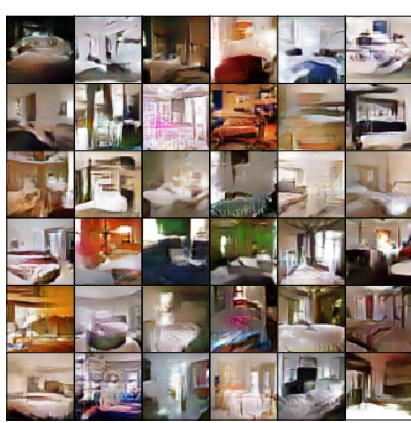 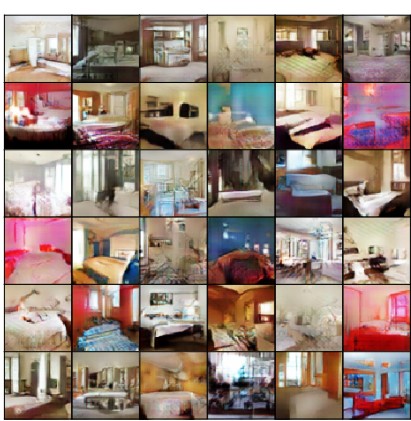 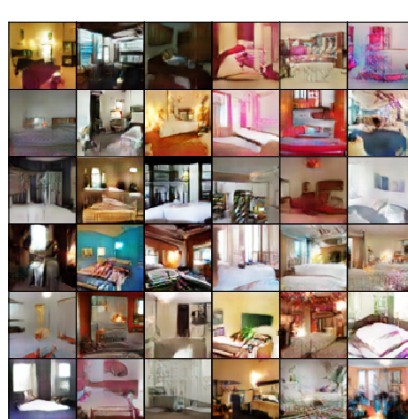

(e) Alternated Extra-Adam

Figure 8: Image generated by RMSProp, Simultaneous and Alternated Extra-Adam, Adam, and Mirror-GAN on the `LSUN bedroom` dataset.

