# OpenReview forum: "Finding Mixed Nash Equilibria of Generative Adversarial Networks"
_ICLR.cc/2019/Conference_

### Official Review · AnonReviewer1 · 2018-11-02
**An interesting mixed strategy perspective to train GANs**

**Rating:** 6
**Confidence:** 4

**Review:**

This paper uses a mixed strategy perspective for GANs. With this formulation the non-convex game formulation of GANs can be transformed into a infinite dimensional problem analog to a finite dimensional bilinear problem.

I really like this approach, that tries to find methods that converge globally to (mixed) Nash equilibriums. However I have some concerns.

- I'm concerned about the definition of a $O(T^{-1})-NE$. Actually, this merit function is not standard for game. It can be 0 even if $x_t,y_t$ is far from the equilibrium (for instance for the problem $\min_{x \in \Delta_d}\max_{y \in \Delta_d} x^\top y$ with $x_t = (1,0,\ldots,0)$ and $y_t= (F(x_{NE},y_{NE}),1-F(x_{NE},y_{NE}),0,\ldots,0)$ we have $F(x_t,y_t) = F(x_{NE},y_{NE})$ but $x_{NE} = y_{NE} =(1/d,\ldots,1/d)$). One merit function that could be considered is $\max_{y} F(x,y_t) - \min_{y} F(x_t,y)$.

- There is a gap between the theory and the practical method that could be bridged. Actually Theorem 2 assume that the stochastic derivatives are unbiased but since your Langevin dynamics gives you an *approximate* of the next distribution an analysis taking into account this bias would provide much stronger results. More precisely, it would be interesting to have a result similar as Theorem 2 with conditions on $\epsilon_t$ and $K_t$. For instance, if the theoretical $K_t$ is too large it would reduce the interest of your algorithm. I think this analysis is key since it allows to claim that you can properly approximate the distributions of interest.

If you are able to ease these concerns I'm eager to increase my grade.


- "(5) is exactly the infinite-dimensional analogue of (1):" Actually it is not exactly the analogue since $<.,.>$ is not a scalar product anymore (particularly, $<g,\mu>$ is not defined) but the canonical pairing between a space and its dual (we are loosing something going to infinite dimension).
I think it should be clarified somewhere.

Minor comments:
- on the updates rules of $\theta$ and $\omega$ (Page 6) the Gaussian noises are missing.
- On algorithm 3,4,5 and 6 the Gaussian noise is too wide and causes an Overfull.

=== After Authors response ===
The authors fixed some major issues. That is why I improved my grade.
However I'm still concerned about the scalability of this algorithm

---

> ### Author Response · Authors · 2018-11-13
> **We thank you for the insightful comments, in particular for suggesting the bias analysis**
>
> Minor comments are incorporated in the revision.
>
> “ I'm concerned about the definition of a $O(T^{-1})-NE$...”
>
> Thank you for noting this; we have fixed the definition. As Reviewer3 mentioned, This was an oversight of the definition, and our convergence guarantees are for the correct definition.
>
>
> “...an analysis taking into account this bias would provide much stronger results.”
>
> Thank you for the very illuminating idea; we now have bounds that explicitly take the bias into account in Theorem 2.
>
> However, we note that quantifying conditions on eps_t and K_t is equivalent to proving non-asymptotic bounds for sampling from distributions over neural nets, a task that is more difficult than proving convergence to global optima, and hence is well beyond the scope of our work. However, under fairly mild conditions, it is known that at least asymptotically, SGLD converges at rate t^{-1/3}; see [*].
>
> “... Actually it is not exactly the analogue since $<.,.>$ is not a scalar product…
>
> Thanks for pointing this out. We have added a footnote to clarify this point.
>
> [*] Consistency and fluctuations for stochastic gradient Langevin dynamics by Teh et al., JMLR 2016.

---

> > ### Comment · AnonReviewer1 · 2018-12-04
> > **On bias analysis**
> >
> > I think that you could bet a more interpretable result in Theorem 2. If you have a bias depending on the time $t$ you could combine it with standard SGLD convergence results (under mild conditions) to get a more practical convergence result. Actually even in the constant bias case we can notice that for a know horizon $T$ the number of *inner* iteration of SGLD to get a O(T^{-1/2})-NE for Algorithm 2 is O(T^{3/2}), making the algorithm potentially not scalable.

---

> > > ### Author Response · Authors · 2018-12-06
> > > **Thanks for the suggestion**
> > >
> > > We would like to note that
> > >
> > > 1. This is the first of its kind convergence result under mild assumptions.
> > >
> > > 2.The main scalability bottleneck of the algorithm is actually the growth of the samples, which we handle via the mean approximation.
> > >
> > >
> > > As a result, we can consider sharpening the analysis if the reviewer insists but the utility of this result feels limited. In practice, preconditioning via adaptive sampling schemes, such as the ones we exploited in the experiments, seem to obviate a more refined analysis. Note that our theory remains intact as any sampling scheme can in principle be used, not only SGLD.
> > >
> > > Finally, we would like to sincerely ask the reviewer to re-evaluate the score, and also encourage the reviewer to engage in a discussion with the other reviewers as well.

---

### Official Review · AnonReviewer3 · 2018-11-03
**interesting extension of mirror-prox with some important missing pieces**

**Rating:** 5
**Confidence:** 5

**Review:**

This paper extends the mirror-descent and mirror-prox algorithms to infinite dimensional Banach spaces so that they can be applied to solve the mixed Nash equilibrium of the popular generative adversarial networks. The main technical results appear to be formal but straightforward extensions of existing techniques in finite dimensional spaces. A sample-based practical algorithm is proposed so that the infinite dimensional algorithms can still be computed. Experiments are a bit disappointing as the authors only used visual appeal as an evaluation criterion. (I understand why the authors chose to do so but as an algorithmic paper, resorting to an evaluation based on visual appeal is almost always unsatisfactory.)

Quality: The quality of this work is moderate. Quite strangely, the authors made a fundamental mistake at the very beginning: their definition of approximate mixed equilibrium  (page 2, Notation) is bizarre and different from those in previous work (such as Nemirovski's MP paper). Fortunately, this is perhaps only an oversight on the definition; the algorithms and theorems are for the correct definition anyways. Example: consider min_{-1<= x <= 1} max_{-1<=y<=1} xy. Should we call (x, 0) an (approximate) NE for any x??

Another major issue with this work is its relaxation into mixed NE. The "bilinarization" trick in Eq (5) goes back to Kantorovich (who perhaps deserves to be mentioned), and is a relaxation in general: we now have to use a mixture of generators. Since MD/MP is not sparse, in the end we must use a large number of mixtures of generators. This certainly will create some computational issues, and make comparison to pure NE methods unfair.

Clarity: The writing of this work is mostly easily to follow. However, the presentation of the technical results suffers from a real dilemma: On one hand, the authors completely ignored the technical difference between infinite dimensional Banach spaces and finite dimensional spaces. In fact, the authors never even formally defined the underlying Banach spaces. Another example, is the mapping G on page 3 continuous? wrt what topology? without such discussion what do you mean by Frechet derivative on page 4? when is the entropy function well-defined? when is the integral of exponential well-defined? Part of me totally understand that these technicalities are daunting and perhaps should not appear in the main text. On the other hand, aren't these technicalities the only "interesting and nontrivial" part of the extension to infinite dimensional spaces? If we do not care about such technicalities and can safely "assume they can be taken care of," then why is this work nontrivial? I do not see a way to resolve this dilemma here but suggest the authors consider maybe a different venue for such type of results.

Originality: The novelty of this work is limited. The extension of MD/MP to infinite dimensional spaces is mostly formal but straightforward. In fact I believe previous authors such as Nemirovski deliberately restrict to finite dimensional spaces not because of technical incapability but to avoid uninspiring technicalities. Some very related previous works were not mentioned at all:
-- Mirror Descent Learning in Continuous Games
-- Convex Games in Banach Spaces
-- On the Universality of Online Mirror Descent

The sample based algorithms are more interesting because they make the infinite dimensional extensions implementable. However, one can not say much about their convergence behavior at the moment.

Significance: The main results, although not difficult to obtain, can potentially be very useful in broadening our arsenal of tools for training GANs. The claim "resolving the longstanding problem that no provably convergent algorithm exists for general GAN" in the Abstract is disturbing, because the authors changed the definition of GAN and because the technical contributions of this work do not live up to that strong claim.

---

> ### Author Response · Authors · 2018-11-13
> **We thank you for the knowledgeable review**
>
> Comments that were already addressed in the general response are omitted. We are more than happy to engage in any further discussion regarding below.
>
> “... the authors made a fundamental mistake …”
>
> Thank you for pointing this out; we have fixed it.
>
> “The "bilinarization" trick in Eq (5) goes back to Kantorovich…”
>
> You might be referring to the Kantorovich’s Duality (KD) for relaxed Monge’s problem. It seems that Kantorovich himself never used any such trick when he derived KD in the classic “On the translocation of masses 1942”. Instead, he directly started with the relaxed problem and used a limit argument. Another potentially related paper is “On a problem of Monge 1948”, which we can only find the Russian version and therefore are unable to check. As a result, we would appreciate if the reviewer can further clarify upon this comment.
>
> In any case, we agree with the reviewer that it is worth mentioning KD; see the new first paragraph in Section 2.2.
>
> “Since MD/MP is not sparse, in the end we must use a large number of mixtures of generators.”
>
> This is precisely why we resort to the averaging scheme; see Section 4.2.
>
>
> “This certainly will create some computational issues, and make comparison to pure NE methods unfair.”
>
> With our averaging scheme, there are no computational issues.
>
> Our comparison to pure NE methods is entirely fair since we are comparing the final output images under similar computational resources, which is ultimately what people care about for GANs.
>
>
> “Clarity: The writing of this work …”
>
> This paragraph warrants a sentence-by-sentence reply; we have addressed all your concerns in the revision.
>
> “”On one hand, the authors completely ignored the technical...””
>
> This comment is perhaps unfair, as we have spent a full section in Appendix A on technical details of inf-dim MD. Even though the presentation in the main text is heuristic, we did refer the readers to Appendix A for precise statements. We are sure that the purpose is well-understood by the reviewer.
>
> “In fact, the authors never even formally defined the underlying Banach spaces.”
>
> We purposely avoided the term “Banach space” for general audience, but an expert should find no difficulty in checking the details of Appendix A. It is easier to start with Banach spaces, but would make the paper less accessible. We are willing to change the presentation if the reviewer feels that the term is necessary.
>
> “”is the mapping G on page 3 continuous ... when is the integral of exponential well-defined?””
>
> Thanks for pointing out these missing pieces. We have incorporated them all with two new paragraphs in Appendix A.
>
> “”Part of me totally understand ... If we do not care about such technicalities and can safely "assume they can be taken care of," then why is this work nontrivial?””
>
> As explained in the general response, we have never claimed that the inf-dim MD is the main contribution. We also handled all the technical details in Appendix A which, thanks to your review, have become more complete in the revision (we also fixed an issue in the definition of our function class).
>
>
> “Originality: The novelty of this work is limited... In fact I believe previous authors such as Nemirovski deliberately restrict to finite dimensional spaces ... ”
>
> Let us reiterate that the technicality is not the major goal of our paper, and we knew that inf-dim MD is folklore among experts; see general response.
>
> “The sample based algorithms are more interesting ... However, one can not say much about their convergence behavior at the moment.”
>
> Please see the general response above; our framework provides, to our knowledge, the strongest theoretical claim for training GANs. For the efficient algorithms with averaging, we did admit that it is heuristic, but it works well in practice and is derived on the guidance provided by the theoretical foundations.
>
> “The claim ... is disturbing, because the authors changed the definition of GAN ...”
>
> We agree that the sentence can be misleading and we have removed it. But we argue that the GAN framework is not married to the classical pure strategy formulation so it is perfectly fine to change the definition of GAN.

---

> > ### Comment · AnonReviewer3 · 2018-12-02
> > **response to the authors**
> >
> > I appreciate the authors' response and I provide some further comments below.
> >
> > “On a problem of Monge 1948:” available here
> > https://link.springer.com/article/10.1007%2Fs10958-006-0050-9
> > (I do not mean you need to cite these papers of Kantorovich, but that you need to provide some history on the linear programming relaxation in infinite dimensional spaces: This is a well-known idea, and perhaps attributing to Kantorovich is appropriate, or someone else if you deem more appropriate.)
> >
> > In Introduction, "with the following contributions: ... 2. We demonstrate that the prox methods of (Nemirovsky & Yudin, 1983; Nemirovski, 2004), which are fundamental building blocks for solving two-player games with finitely many strategies, can be extended to continuously many strategies, and hence applicable to training GANs. We provide an elementary proof for their convergence rates to learning the mixed NE."
> >
> > In Paragraph 3 "to our knowledge, these results are new."  And then you supply two theorems in Section 3 for the infinite-dimensional MD whereas no theorems were provided for the sampling algorithm later. I hope you can see why the reviewers (misleadingly) thought you were making a big deal of this part.
> >
> > "We purposely avoided the term “Banach space” for general audience, but an expert should find no difficulty in checking the details of Appendix A. It is easier to start with Banach spaces, but would make the paper less accessible."
> >
> > I perfectly understand the dilemma here. My point is you either reduce Section 3 to its minimum (and defer the technicalities entirely to the appendix) or you go completely rigorous. I see no point in being vague and imprecise in the main text: for a general reader many of the notions are just difficult to comprehend while for an expert your notions are full of impreciseness. (Another example: your definition of Frechet derivative is wrong. It should be the Gateaux derivative.) Presenting in the current form has the risk of annoying every possible reader.
> >
> > I should add that I totally agree that there is definitely value in formally stating and proving the convergence of infinite-dimensional MD (as I stated in the original review). It is just perhaps the wrong venue. (Don't you want a qualified reviewer to properly check your appendix? ICLR reviewers, due to the load, are not obliged to do so, as you must be fully aware.)
> >
> > "we argue that the GAN framework is not married to the classical pure strategy formulation so it is perfectly fine to change the definition of GAN."
> >
> > Totally. You can change the definition of GAN but you just do not claim by doing so you resolve a longstanding open problem (unless you prove the relaxation is tight), and you do not call the new definition as GAN again.
> >
> > Lastly, another issue that the authors might have overlooked: the formulation of GAN (such as minimizing f-divergence or the Wasserstein distance) is actually well-defined and admits a unique solution. The difficulty only comes in when we use a nonconvex neural network to do certain approximations (as the authors pointed out in the  response). Now we have defined a mixed-equilibrium formulation (which is well-defined even with a nonconvex neural network approximation), but due to infinite-dimension we have to do some computational approximation. Does this sound familiar? How do we know if the approximate MD would behave benignly?

---

> > > ### Author Response · Authors · 2018-12-04
> > > **Thank you for the constructive response**
> > >
> > > “On a problem of Monge ....”
> > >
> > > We thank you for the translation reference. We have already mentioned in the revision that the technique is useful in mathematics and we pointed out its value in optimal transport.
> > >
> > > “In Introduction .... I hope you can see why the reviewers (misleadingly) thought you were making a big deal of this part. “
> > >
> > > We respectfully disagree with your comment. We explicitly state four contributions in the introduction. Based on the rebuttal, hopefully we can convince the reviewers to evaluate the other three as well.
> > >
> > > In particular for ICLR, a new perspective and the end-to-end engineering are important. Our contributions in proposing novel problem formulations and also taking care of the non-trivial implementation should not be underestimated, given it outperforms methods which search only pure strategy equilibria.
> > >
> > > “I perfectly understand the dilemma here…”
> > >
> > >
> > > We had already minimized Section 3: It is only one-page long. It is accessible to the general audience while the appendix supplements the precise derivations.
> > >
> > > We would be happy if the reviewer can provide further suggestions about improving presentation as you did in the initial comments. However, we also contend that the reviewer’s arguments so far have not entirely justified the low score.
> > >
> > > Thank you very much for pointing out the glitch with Frechet derivative; we note that entropy is NOT Gateau differentiable but it possesses Frechet derivatives at all the directions in the definition of our function class; see Appendix A.
> > >
> > > “I should add that I totally agree that there is definitely value in formally stating and proving the convergence of infinite-dimensional MD (as I stated in the original review). It is just perhaps the wrong venue. (Don't you want a qualified reviewer to properly check your appendix? ICLR reviewers, due to the load, are not obliged to do so, as you must be fully aware.)”
> > >
> > > In our opinion, the technical level in this paper does not meet the level of a premier theory venue. However, there is clear practical value in our work thanks to the theoretical guidance. Philosophically, we also do not think that it is reviewers’ job to examine the proof in its entirety; it is our job to ensure the correctness of the paper.
> > >
> > > At the same time, experienced reviewers can always grasp high-level messages and judge accordingly.  To be more fair, we sincerely ask you to concentrate less on the inf-dim MD part (which is only about one page in the main text) and re-evaluate the our contributions based on the revised full paper and also our discussion for ICLR.
> > >
> > > “Totally. You can change the definition of GAN... “
> > >
> > >
> > > The term “GAN” for us is a general concept representing the training paradigm where an auxiliary adversarial party is added. We understand that there is not a universal agreement so we have removed the sentence in the revision.
> > >
> > > “Lastly, another issue that the authors might have overlooked: the formulation of GAN (such as minimizing f-divergence or the Wasserstein distance) is actually well-defined and admits a unique solution. The difficulty only comes in when we use a nonconvex neural network to do certain approximations (as the authors pointed out in the response).”
> > >
> > > The formulation of GAN is well-defined on the level of probability measures and functions but NOT on the level when we parametrize them with neural networks. Our framework IS well-defined even when neural networks are involved. We have never overlooked this; in fact, we see this as the major advantage for solving mixed Nash Equilibrium: it is always well-defined no matter what parametric models you use, and this is simply wrong for the pure strategy equilibrium.
> > >
> > > “Now we have defined a mixed-equilibrium formulation (which is well-defined even with a nonconvex neural network approximation), but due to infinite-dimension we have to do some computational approximation. Does this sound familiar? How do we know if the approximate MD would behave benignly?”
> > >
> > > We again disagree with the reviewer. We justify the approximation by examining them with numerical evidence. We conducted experiments in the paper, and the main routine (sampling) was also reported successful in the many previous studies that we cited. We also stress again that, assuming SGD/Adam/etc. gives you the solution of the minimization problem, there is currently nothing we can say about optimizing GANs.
> > >
> > > If the reviewer rejects our paper based on the argument that the computational approximation is not a good option, then this would imply that all the papers that motivate from convex optimization perspective and then heuristically deploy them to neural nets have low research value. Such a track is nonetheless the mainstream of state-of-the-art understanding of training GANs (see also the concurrent submission cited in our previous rebuttal), and has lead to many valuable insights. In our opinion, making valid assumptions should be valued, instead of dismissed.

---

### Official Review · AnonReviewer2 · 2018-11-05
**Generalize mirror descent to infinite dimensional spaces. No really new theory or practice insight.**

**Rating:** 4
**Confidence:** 4

**Review:**

This paper proposes to consider the mixed equilibrium objective function for GANS. The authors generalize the mirror descent/mirror prox to handle continuous games. The technical challenge is to write those algorithms in infinite dimensional spaces. This reviewer finds this however to be a mere technicality, and there seems to be no conceptual obstruction. In fact other paper have already written this, see for example ``Mirror Descent Learning in Continuous Games" by Zhou et al. at CDC 2017 (I'm sure there are other references too).
While the theory part is not particularly exciting, the paper could be saved by the experiments. However as far I can tell the authors are only able to reproduce the results obtained with more classical approaches.

---

> ### Author Response · Authors · 2018-11-13
> **Thank you for your comments**
>
> We have addressed all of your concerns and we look forward to further correspondence with you.
>
> “ The technical challenge is to write those algorithms in infinite dimensional spaces. This reviewer finds this however to be a mere technicality, and there seems to be no conceptual obstruction.”
>
> Please see the ****Novelty and significance**** part of the general response.
>
> The conceptual leap here is not the infinite dimensional algorithm; rather, it is the infinite dimensional re-formulation of the GAN problem via the Riesz representation.
>
> We do know the entropic mirror descent solves the problem in finite dimension, however it is surprising to see that the mirror-prox along with Langevin dynamics handle the GAN problem in a simple fashion, which this paper brings to the table. Our results are not some theory dressing to motivate a method that already works in practice.
>
> Our formulation, the algorithm (a fusion of entropic mirror descent and Langevin dynamics), and the practical results, to our knowledge, is a solid contribution to the literature. Without the infinite dimensional characterization, we felt like we would be criticized for lack of rigor.
>
> “ In fact other paper have already written this, see for example ``Mirror Descent Learning in Continuous Games’’ by Zhou et al. at CDC 2017 (I'm sure there are other references too).”
>
> The paper, as explained in the general response, is not relevant to this work. Regarding algorithms for solving infinite-dimensional games, the reference (Balandat et al., 2016) is the most appropriate to our knowledge. We would appreciate if you can point out some references that are closer to our work than (Grnarova et al. 2018) or (Balandat et al., 2016), which we might have possibly missed.
>
>
> “While the theory part is not particularly exciting, the paper could be saved by the experiments. However as far I can tell the authors are only able to reproduce the results obtained with more classical approaches.”
>
> We now have a comparison to the most contemporary algorithm; please see the general response above and the revision.

---

### Author Response · Authors · 2018-11-13
**General response: ****Novelty and significance******

Reviewer2 and 3 criticized our work from lacking sufficient technicality on infinite-dimensional mirror descent (inf-dim MD); we argue that this is an oversight of our contribution. It is also not a fair evaluation.

Simply put, prior to our work, the literature treats the GAN formulations as if they are (semi) convex-concave games. Then, you see familiar optimization methods from the deep, earlier literature in game theory and online learning applied to the GAN problem, basically ignoring the non-convexity of neural nets (except for the very few literature that we cited).

Our work proposes a new insight, which is obvious in retrospect for theorists: We seek distributions over GAN parameters and show that the underlying learning problem is a well-posed, bi-affine convex game, albeit infinite dimensional.

We build on this insight by showing how the infinite-dimensional entropic MD applies to it with rates while the iterations of this algorithm can be obtained via Langevin dynamics. We also take care of the end-to-end engineering aspects of this proposal and integrate an algorithm that has excellent practical performance.

There is no such work in the literature, which encouraged us to state that we resolved an open problem.

We contend that none of the elements of our approach has theory dressing or obfuscation. We do not use an existing algorithm that has guarantees with a different set of assumptions and yet still apply to GANs. All the elements in our approach are necessary. If you take any one of them out, whether it is the infinite dimensional bi-affine game reformulation via Riesz representation, or Langevin dynamics-based iterations, the framework falls down.

We also know that people technically skilled can do the infinite-dimensional MD extensions, but since there is nothing to cite currently, we put the derivation for completeness.

In fact, Marc Teboulle mentioned to us in private communication in October 2018 that he had the infinite-dimensional characterization of the entropic MD done but did not publish. We are also personally aware that Arkadi Nemirovskii is more than capable of doing such a derivation, along with many other optimization experts.

That being said, however, we would appreciate if the reviewers can point out to us any literature that has the derivation written down and how it connects with Langevin dynamics for sampling. The reason for this terse remark is that if we do not provide the derivation, it is a point for criticism. If we do it for completeness, as in the current paper, it should not be a reason for omitting our real contributions and over-emphasizing as if it is the main contribution of this work in an overly harsh manner.

Finally, we stress that our theory allows to make the strongest theoretical claim for training GANs to date: Assuming the success of sampling from a single neural net, which was empirically justified in previous work, we can show non-asymptotic convergence rates to the saddle points. In contrast, even if we assume that SGD/Adam/etc. globally optimize a single neural net, can we say anything about the convergence to the saddle points of classical GANs? To our knowledge, the answer is currently none.

---

### Author Response · Authors · 2018-11-13
**General response: ****Literature******

As the reviewers are keen on pointing out subtleties in contributions, we would like to highlight that we were very careful with our citations.

First of all, we stress that the paper, which we were aware of at the point of submission,

[1] Mirror Descent Learning in Continuous Games by Zhou et. al. at CDC 2017

is of little relevance to our study. Since [1] assumes a CONCAVE reward, ordinary finite-dimensional mirror descent already solves the PURE strategy equilibrium.

In particular, there is no notion of mixed Nash Equilibrium (mixed NE) in [1], and all the optimization variables are in subsets of R^d, and hence, it is completely different from our setting. When applied to training GANs, one necessarily faces the problem of non-concavity, and no result in [1] can hold. This problem was studied by (Grnarova et al. 2018) in our submission, which we have cited and compared to in detail.

We also stress that we have already cited the most relevant literature (Balandat et al., 2016) on convex games in Banach spaces, and the relevance in other related works (such as the ones provided by Reviewer3) are contained in (Balandat et al., 2016).

Specifically, our framework is algorithmic, and we study convergence of a specific, implementable, algorithm, which has the same theme as (Balandat et al., 2016).

On the other hand, the second and third paper provided by Reviewer3 concern existence of mirror maps and do not provide any algorithm. Moreover, the results in both papers do not apply to our problem.

On one hand, they rely on theory of martingale type which gives nontrivial bounds only for L^p spaces with 1<p<\infty, whereas the special case of p=1 is essential to our interest.

Moreover, the third paper assumes the optimization constraint to be centrally-symmetric, which does not hold for the set of all probability distributions.

Please let us know if we have missed other references.

---

### Author Response · Authors · 2018-11-13
**General response: ****New experiments******

As Reviewer2 suggested, we performed additional numerical comparisons to modern methods. The revision includes two contemporary algorithms, including the simultaneous and alternated Extra-Adam, from the concurrent ICLR submission

[2] A Variational Inequality Perspective on Generative Adversarial Networks.

While the theory of [2] results in an algorithm for simultaneous updates, as in our method, the authors in the numerical evidence use an alternating update scheme, whose convergence guarantee is unknown. Moreover, note also that the theorems in [2] apply only to the simultaneous update algorithm for the (strongly) convex-concave objectives and not the GAN objective. Hence, the alternating update algorithm is mostly motivated by the empirical evidence.

In the new Figure 2, one can see that the simultaneous Extra-Adam fails in our experiment, while the performance of alternated Extra-Adam is comparable to Mirror-GAN. Currently it is unclear with one dataset whether our approach yields better empirical result than [2]. However, we highlight that our algorithm is motivated by a clear mixed NE theoretical model and approximations that directly apply to this model.

Finally, the alternated Extra-Adam slightly outperformed our Mirror-GAN on the LSUN dataset in terms of Inception Score, whereas the slight improvement in the numerics, in our opinion, did not translate into the quality of the generated images; see Table 2. We plan on further testing our method on other datasets as well.

---

### Meta-Review · Area_Chair1 · 2018-12-18
**Revise and resubmit**

**Confidence:** 4
**Recommendation:** Reject

**Metareview:**

While the authors made a strong rebuttal, none of the reviewers were particularly enthusiastic about the contributions of this paper and we unfortunately have to reject borderline papers. Concerns were expressed about the presentation, as well as the scalability of the approach. The AC encourages the authors to "revise and resubmit".